# Random Feature Representation Boosting

**Nikita Zozoulenko** [1]  **Thomas Cass** [* 1]  **Lukas Gonon** [* 1 2]

## Abstract

We introduce Random Feature Representation Boosting (RFRBoost), a novel method for constructing deep residual random feature neural networks (RFNNs) using boosting theory. RFRBoost uses random features at each layer to learn the functional gradient of the network representation, enhancing performance while preserving the convex optimization benefits of RFNNs. In the case of MSE loss, we obtain closed-form solutions to greedy layer-wise boosting with random features. For general loss functions, we show that fitting random feature residual blocks reduces to solving a quadratically constrained least squares problem. Through extensive numerical experiments on tabular datasets for both regression and classification, we show that RFRBoost significantly outperforms RFNNs and end-to-end trained MLP ResNets in the small- to medium-scale regime where RFNNs are typically applied. Moreover, RFRBoost offers substantial computational benefits, and theoretical guarantees stemming from boosting theory.

## 1. Introduction

Random feature neural networks (RFNNs) are single-hidden-layer neural networks where all model parameters are randomly initialized or sampled, with only the linear output layer being trained. This approach presents a computationally efficient alternative to neural networks trained via stochastic gradient descent (SGD), avoiding the challenges associated with non-convex optimization and vanishing/exploding gradients. Despite their simplicity, RFNNs and related random feature models have strong provable generalization guarantees (Rahimi & Recht, 2008b; Rudi & Rosasco, 2017; Lanthaler & Nelsen, 2023; Cheng et al., 2023), and have demonstrated state-of-the-art performance

---
[*]Equal last authors. [1]Department of Mathematics, Imperial College London, UK [2]School of Computer Science, University of St. Gallen, Switzerland. Correspondence to: Nikita Zozoulenko <n.zozoulenko23@imperial.ac.uk>.

*Proceedings of the 42$^{nd}$ International Conference on Machine Learning*, Vancouver, Canada. PMLR 267, 2025. Copyright 2025 by the author(s).

and speed across various tasks (Bolager et al., 2023; Dempster et al., 2023; Gattiglio et al., 2024; Prabhu et al., 2024).

Recent theoretical work on Fourier RFNNs has shown that deep residual RFNNs can achieve lower generalization errors than their single-layer counterparts (Kammonen et al., 2022). Current theory and algorithms for training deep RFNNs, however, are limited to the Fourier activation function (Davis et al., 2024), and uses ideas from control theory to sample from optimal weight distributions. While skip connections have been crucial to the success of deep end-to-end-trained ResNets (He et al., 2015), introducing them into general RFNNs is not straightforward, as naively stacking random layers may degrade performance. In this paper, we introduce **random feature representation boosting (RFRBoost)**, a novel method for constructing deep ResNet RFNNs. Our approach not only significantly improves performance, but also retains the highly tractable convex optimization framework inherent to RFNNs, with theoretical guarantees stemming from boosting theory.

ResNets have traditionally been studied from two primary perspectives. The first views a ResNet $\Phi_t$, defined by

$$\Phi_t(x) = \Phi_{t-1}(x) + g_t(\Phi_{t-1}(x)), \qquad (1)$$

as an Euler discretization of a dynamical system $d\Phi_t(x) = g_t(\Phi_t)dt$ (E, 2017). Here $g_t$ represents a residual block at layer $t$, often expressed as $g_t(x) = A\sigma(Bx+b)$. This framework was later generalized to the setting of neural ODEs (Chen et al., 2018; Dupont et al., 2019; Kidger et al., 2020; 2021; Walker et al., 2024). The second point of view is that of gradient boosting, where a ResNet can be seen as an ensemble of weak, shallow neural networks of varying sizes, $\Phi_T(x) = \sum_{t=1}^T g_t(\Phi_{t-1}(x))$, derived by unravelling equation (1) (Veit et al., 2016). This led to the development of gradient representation boosting (Nitanda & Suzuki, 2018; Suggala et al., 2020), which studies residual blocks via functional gradients in the space of square integrable random vectors $L_2^D(\mu)$, where $\mu$ is the distribution of the data.

A key challenge in extending RFNNs to deep ResNet architectures lies in the crucial role of the residual blocks $g_t$ when they are composed of random features. If the magnitude of $g_t$ is too small, the initial representation $\Phi_0$ dominates, rendering the added random features ineffective. Conversely, if $g_t$ is too large, information from previous layers can be lost. This problem is not merely one of scale; ideally each

residual block should approximate the negative functional gradient of the loss with respect to the network representation. However, random layers are not guaranteed to possess this property. Unlike end-to-end trained networks where SGD with backpropagation can adjust all network weights to learn an appropriate scale and representation, RFNNs lack a comparable mechanism because their hidden layers are fixed. We address this issue by using random features at each layer of the ResNet to learn a mapping to the functional gradient of the training data, enabling tractable learning of optimal random feature residual blocks via analytical solutions or convex optimization.

## 1.1. Contributions

Our paper makes the following contributions:

- **Introducing RFRBoost:** We propose a novel method for constructing deep ResNet RFNNs, overcoming the limitations of naively stacking random features layers. RFRBoost supports arbitrary random features, extending beyond classical random Fourier features.

- **Analytical Solutions and Algorithms:** For MSE loss, we derive closed-form solutions to greedy layer-wise boosting using random features by solving what we term *"sandwiched least squares problems"*, a special case of generalized Sylvester equations. For general losses, we show that fitting random feature residual blocks is equivalent to solving a quadratically constrained least squares problem.

- **Theoretical Guarantees:** We provide a regret bound for RFRBoost based on Rademacher complexities and established results from boosting theory.

- **Empirical Validation:** Through numerical experiments on 91 tabular regression and classification tasks from the curated OpenML repository, we demonstrate that RFRBoost significantly outperforms both single-layer RFNNs and end-to-end trained MLP ResNets, while offering substantial computational advantages.

## 1.2. Related literature

**Classical Boosting:** Boosting aims to build a strong ensemble of weak learners via additive modelling of the objective function, dating back to the 1990s with the development of AdaBoost (Freund & Schapire, 1997). Gradient boosting, introduced as a generalization of boosting, supports general differentiable loss functions (Mason et al., 1999; Friedman et al., 2000; Friedman, 2001), and includes popular frameworks such as XGBoost (Chen & Guestrin, 2016), LightGBM (Ke et al., 2017), and CatBoost (Prokhorenkova et al., 2018). These models typically use decision trees as weak learners and are widely considered the best out-of-the-box models for tabular data (Grinsztajn et al., 2022).

**Boosting for Neural Networks:** Applications of boosting for neural network first appeared in AdaNet (Cortes et al., 2017), which built a network graph using boosting to minimize a data-dependent generalization bound. Huang et al. (2018) introduced an AdaBoost-inspired algorithm for sequentially learning residual blocks, boosting the feature representation rather than the class labels. Nitanda & Suzuki (2018; 2020) proposed using Fréchet derivatives in $L_2^D(\mu)$ to learn residual blocks that preserve small functional gradient norms, motivated by Reproducing Kernel Hilbert Space (RKHS) theory and smoothing techniques. GrowNet (Badirli et al., 2020) constructs an ensemble of neural networks in the classical sense of gradient boosting the labels, not as a ResNet, but by concatenating the features of previous models to the next weak learner. Suggala et al. (2020) introduced *gradient representation boosting*, which studies layer-wise training of ResNets by greedily or gradient-greedily minimizing a risk function, and gives modular excess risk bounds based on Rademacher complexities. Yu et al. (2023) used gradient boosting to improve the training of dynamic depth neural networks. Finally, Emami & Martínez-Muñoz (2023) constructed a network neuron-by-neuron using gradient boosting.

**Random feature models:** Random feature models use fixed, randomly generated features to map data into a high dimensional space, enabling efficient linear learning. These methods encompass a broad class of models studied under various names, including random Fourier features (Rahimi & Recht, 2007; Sriperumbudur & Szabo, 2015; Li et al., 2019; Kammonen et al., 2022; Davis et al., 2024), extreme learning machines (Huang et al., 2004; 2012; Huang, 2014), random features or RFNNs (Huang et al., 2006; Rahimi & Recht, 2008a;b; Rudi & Rosasco, 2017; Carratino et al., 2018; Yehudai & Shamir, 2019; Mei & Montanari, 2022; Bolager et al., 2023; Lanthaler & Nelsen, 2023; Ayme et al., 2024), reservoir computing (Jaeger, 2001; Lukoševičius & Jaeger, 2009; Gallicchio et al., 2017; Grigoryeva & Ortega, 2018; Tanaka et al., 2019; Hart et al., 2020; Gonon et al., 2023; 2024), kernel methods (Kar & Karnick, 2012; Sinha & Duchi, 2016; Sun et al., 2018; Szabo & Sriperumbudur, 2019; Cheng et al., 2023; Wang et al., 2024; Wang & Feng, 2024), and scattering networks (Bruna & Mallat, 2013; Cotter & Kingsbury, 2017; Oyallon et al., 2019; Trockman et al., 2023). Recently, RFNNs and related approaches have demonstrated exceptional performance in both speed and accuracy across a wide variety of applications. These include solving partial differential equations (Nelsen & Stuart, 2021; Gonon, 2023; Gattiglio et al., 2024; Neufeld & Schmocker, 2024), online learning (Prabhu et al., 2024), time series classification (Dempster et al., 2020; 2023), and in mathematical finance (Jacquier & Zuric, 2023; Herrera et al., 2024). RFNNs have also been explored in the context of quantum computing (Innocenti et al., 2023; Gonon &

Jacquier, 2023; Martínez-Peña & Ortega, 2023; Xiong et al., 2024), and in relation to random neural controlled differential equations and state-space models (Cuchiero et al., 2021; Cirone et al., 2023; 2024; Biagini et al., 2024).

# 2. Functional Gradient Boosting

In this section we introduce notation and give a brief overview of classical gradient boosting (Mason et al., 1999; Friedman, 2001), and its connection to deep ResNets via gradient representation boosting (Nitanda & Suzuki, 2018; 2020; Suggala et al., 2020).

Let $(X, Y) \sim \mu$ be a random sample with its corresponding probability measure $\mu$. Denote by $X \sim \mu_X$ the features in $\mathbb{R}^q$, with targets $Y \sim \mu_Y$ in $\mathbb{R}^d$, and $\mu_{Y|X=x}$ the conditional law. Let $\widehat{\mu} = \frac{1}{n} \sum_{i=1}^{n} \delta_{(x_i, y_i)}$ be the empirical measure of $\mu$. We will work in the Hilbert space $L_2^D(\mu)$, with inner product given by $\langle f, g \rangle_{L_2^D(\mu)} = \mathbb{E}_\mu[\langle f, g \rangle_{\mathbb{R}^D}]$. The concept of functional gradient plays a key role in gradient boosting:

**Definition 2.1.** Let $\mathscr{H}$ be a Hilbert space. A function $f : \mathscr{H} \to \mathbb{R}$ is said to be **Frechet differentiable** at $h \in \mathscr{H}$ if there exists a bounded linear operator $\nabla f : U \to \mathscr{H}$ for some open neighbourhood $U \subset \mathscr{H}$ of $h$ such that

$$f(h + g) = f(h) + \langle g, \nabla f(h) \rangle_{\mathscr{H}} + o(\|g\|_{\mathscr{H}})$$

for $g \in U$. The element $\nabla f(h)$ is often referred to as the functional gradient of $f$ at $h$, or simply the gradient.

## 2.1. Traditional Gradient Boosting

Let $l : \mathbb{R}^d \times \mathbb{R}^d \to \mathbb{R}$ be a sufficiently regular loss function. Traditional gradient boosting seeks to minimize the risk $\mathcal{R}(F) = \mathbb{E}_\mu[l(F(X), Y)]$ by additively modelling $F \in \text{span}(\mathcal{G})$ within a space of weak learners $\mathcal{G}$, typically a set of decision trees (Friedman, 2001; Chen & Guestrin, 2016). It iteratively updates an estimate $F_t = \sum_{s=1}^{t} \eta \alpha_s g_s$ based on a first-order expansion of the risk:

$$\mathcal{R}(F_t + g) \approx \mathcal{R}(F_t) + \langle g, \nabla \mathcal{R}(F_t) \rangle_{L_2^d(\mu)}.$$

To minimize the risk based on this approximation, a new weak learner $g_{t+1}$ is fit to the negative functional gradient, $-\nabla \mathcal{R}(F_t)$, and the ensemble is then updated: $F_{t+1} = F_t + \eta \alpha_{t+1} g_{t+1}$, where $\eta, \alpha_{t+1} > 0$ are the global and local learning rates. The functional gradient can be shown to be equal to

$$\nabla \mathcal{R}(F)(x) = \mathbb{E}_{\mu_{Y|X=x}}[\partial_1 l(F(x), Y)],$$

and is easily computed for empirical measures $\widehat{\mu} = \frac{1}{n} \sum_{i=1}^{n} \delta_{(x_i, y_i)}$ using the empirical risk $\widehat{\mathcal{R}}(F)$. The method for fitting $g_{t+1}$ varies between implementations and depends on the class of weak learners, as seen in modern approaches like XGBoost (Chen & Guestrin, 2016) which incorporates second-order information.

## 2.2. Gradient Representation Boosting

Unlike classical boosting, which boosts in target space, neural network gradient representation boosting aims to additively learn a feature representation by modelling $F$ via $F = F_t(X) = W_t^\top \Phi_t(X)$, where $\Phi_t = \sum_{s=0}^{t} \eta g_s$ is a gradient boosted feature representation and $W_t \in \mathbb{R}^{D \times d}$ is the top-level linear predictor. In other words, gradient representation boosting seeks to learn the feature representation $\Phi_t$ additively to build a single strong predictor, rather than constructing $F_t$ as an ensemble of weak predictors (Huang et al., 2018; Nitanda & Suzuki, 2018; 2020; Suggala et al., 2020). For a depth $t$ ResNet $\Phi_t$, defined recursively as

$$\Phi_t = \Phi_{t-1} + \eta h_t(\Phi_{t-1}), \qquad (2)$$

we see by unravelling (2), that the weak learners $g_t$ are of the form $g_t = h_t(\Phi_{t-1})$. These functions $g_t$ are sometimes referred to as *weak feature transformations*. Prior work has employed shallow neural networks as $g_t$, trained greedily layer-by-layer with SGD (see references above). In the setting of gradient representation boosting we denote the risk as $\mathcal{R}(W, \Phi) = \mathbb{E}_\mu[l(W^\top \Phi(X), Y)]$, and the functional gradient with respect to the feature representation $\Phi$ is

$$\nabla_2 \mathcal{R}(W, \Phi)(x) = \mathbb{E}_{\mu_{Y|X=x}}[W \partial_1 l(W^\top \Phi(X), Y)].$$

Let $\widehat{\mathcal{R}}(W, \Phi)$ be the empirical risk, $\mathcal{W}$ the hypothesis set of top-level linear predictors, and $\mathcal{G}_t$ the set of weak feature transformations. As outlined by Suggala et al. (2020), there are two main strategies for training ResNets via gradient representation boosting, which we describe below.

**1.) Exact-Greedy Layer-wise Boosting:** At boosting step $t$, the model is updated greedily by solving the joint optimization problem

$$W_t, g_t = \underset{W \in \mathcal{W}, g \in \mathcal{G}_t}{\operatorname{argmin}} \widehat{\mathcal{R}}(W, \Phi_{t-1} + g), \qquad (3)$$

leading to the update $\Phi_t = \Phi_{t-1} + \eta g_t$. This approach is feasible when $W$ and $g$ are jointly optimized by SGD. For general ResNets, this translates to $g_t = h_t(\Phi_{t-1})$, where $h_t$ is a shallow neural network (i.e., a residual block) and $W$ the top-level linear predictor.

However, in the context of random feature ResNets, using SGD for the joint optimization in Equation (3) undermines the computational benefits inherent to the random feature approach. As will be demonstrated in Section 3, by restricting $g_t$ to a simple residual block $g_t = A_t f_t(\Phi_{t-1})$, with $f_t$ representing random features and $A_t$ a linear map, we can derive closed-form analytical solutions for a two-stage approach to the optimization problem (3). This speeds up training and removes the need for hyperparameter tuning of the scale of the random features $f_t$ at each individual layer.

**2.) Gradient-Greedy Layer-wise Boosting:** An alternative to directly minimizing the risk by solving (3), which can be

intractable depending on the loss $l$ and the family of weak learners, is to follow the negative functional gradient. At boosting iteration $t$, this approach approximates the risk using the first-order functional Taylor expansion:

$$\mathcal{R}(W, \Phi + g) \approx \mathcal{R}(W, \Phi) + \langle g, \nabla_2 \mathcal{R}(W, \Phi)\rangle_{L_2^D(\mu)}. \quad (4)$$

A new weak learner $g_t \in \mathcal{G}_t$ is fit by minimizing the empirical inner product $\langle g, \nabla_\Phi \widehat{\mathcal{R}}(W_{t-1}, \Phi_{t-1})\rangle_{L_2^D(\widehat{\mu})}$. The depth $t$ feature representation is obtained by $\Phi_t = \Phi_{t-1} + \eta g_t$, and the top-level predictor is then updated:

$$W_t = \underset{W \in \mathcal{W}}{\operatorname{argmin}} \widehat{\mathcal{R}}(W, \Phi_t).$$

**Potential Challenges:** The gradient-greedy approach for constructing ResNets has remained relatively unexplored in the literature. Suggala et al. (2020), who introduced both strategies in the context of end-to-end trained networks, only mention in passing that the gradient-greedy strategy performed worse comparatively. We believe this might be attributed to the functional direction of $g$ failing to be properly preserved during training with SGD. Specifically, when $g$ is chosen from the family $g = h(\Phi_{t-1})$, where $h$ is a shallow neural network, minimizing $\langle g, \nabla_2 \widehat{\mathcal{R}}(W_{t-1}, \Phi_{t-1})\rangle_{L_2^D(\widehat{\mu})}$ via SGD might increase $\|g\|_{L_2^D(\widehat{\mu})}$ without ensuring that $g$ aligns with the functional gradient in $L_2^D(\widehat{\mu})$. Since the first-order approximation (4) only holds for $g$ with small functional norm, we argue that this objective should instead be minimized under the constraint $\|g\|_{L_2^D(\widehat{\mu})} = 1$. In our random feature setting, we incorporate this constraint and show in Section 4 that it leads to the gradient-greedy approach outperforming the exact-greedy strategy.

# 3. Random Feature Representation Boosting

Our goal is to construct a random feature ResNet of the form

$$\Phi_t = \Phi_{t-1} + \eta A_t f_t,$$

where $f_t = f_t(x, \Phi_{t-1}(x)) \in \mathbb{R}^p$ are random features, $A_t \in \mathbb{R}^{D \times p}$ is a linear map, $x$ is input data, and $\eta > 0$ is the learning rate. We propose to use the theory of gradient representation boosting (Suggala et al., 2020) to derive optimal expressions for $A_t$. We consider three different cases where $A_t$ is either a scalar (learning optimal learning rate), a diagonal matrix (learning dimension-wise learning rate), or a dense matrix (learning the functional gradient). Note that the scalar and diagonal cases assume that $p = D$.

This section is outlined as follows: We first define a random feature layer. We then analyze the case of MSE loss $l(x, y) = \frac{1}{2}\|x - y\|_{\mathbb{R}^d}^2$, deriving closed-form solutions for layer-wise exact greedy boosting with random features. We then explore the gradient-greedy approach, which supports any differentiable loss function.

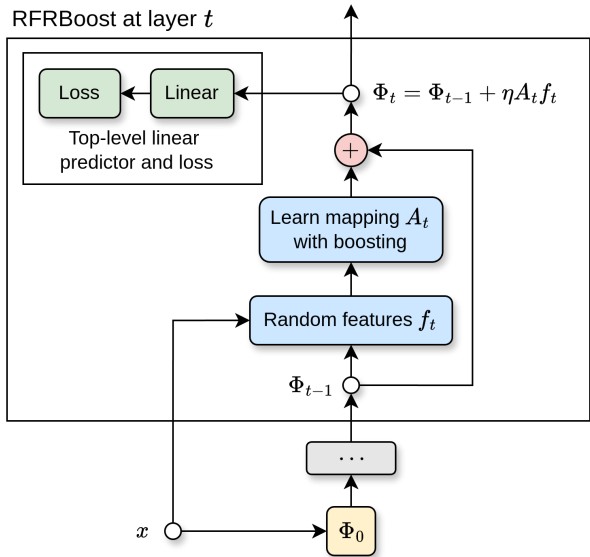

RFRBoost at layer $t$

*Figure 1.* Diagram of random feature representation boosting.

## 3.1. Random Feature Layer

With a random feature layer, we mean any mapping which produces a feature vector $f_t \in \mathbb{R}^p$, which will not be trained after initialization. This is in contrast to end-to-end trained networks, where all model weights are adjusted continuously during training. A common choice for $f_t$ is a randomly sampled dense layer, $f_t(x) = \sigma(B_t \Phi_{t-1}(x))$, with activation function $\sigma$ and weight matrix $B_t \in \mathbb{R}^{p \times D}$. To enhance the expressive power of RFRBoost, we allow the random features $f_t$ to be functions of both the input data $x$ and the previous ResNet layer's output $\Phi_{t-1}(x)$, similar to GrowNet (Badirli et al., 2020) and ResFGB-FW (Nitanda & Suzuki, 2020). Figure 1 provides a visual representation of the RFRBoost architecture. Specifically, in our experiments on tabular data, we let the random feature layer be $f_t(x) = \sigma(\operatorname{concat}(B_t \Phi_{t-1}(x), C_t x))$. The initial mapping $\Phi_0$ can for instance be a random fixed linear projection, or the identity. Instead of initializing $B_t$ and $C_t$ i.i.d., we use SWIM random features (Bolager et al., 2023) in our experiments (see Appendix E for more details). Note however that RFRBoost supports arbitrary types of random features, not only randomly initialized or sampled dense layers.

## 3.2. Exact-Greedy Case for Mean Squared Error Loss

Recall that in the exact-greedy layer-wise paradigm, the objective at layer $t$ is to find a function $g_t \in \mathcal{G}_t$ that additively and greedily minimizes the empirical risk $\widehat{\mathcal{R}}(W_t, \Phi_{t-1} + g_t)$ for some linear map $W_t \in \mathcal{W}$, given a ResNet feature representation $\Phi_{t-1}$. We propose using functional gradient boosting to construct a residual block of the form $g_t = A_t f_t$, where $A_t \in \mathbb{R}^{D \times p}$ is a linear map, and $f_t$ is a random feature layer. We consider the cases where $A_t$ is a scalar

multiple of the identity, a diagonal matrix, or a general dense matrix. The procedure is outlined below:

**Step 1:** Generate random features $f_t = f_t(x, \Phi_{t-1}(x))$, which may depend on both the raw training data $x$ and the activations at the previous layer $\Phi_{t-1}(x)$. Using MSE loss, we find that

$$A_t = \underset{A}{\operatorname{argmin}} \widehat{\mathcal{R}}(W_{t-1}, \Phi_{t-1} + Af_t(x_i))$$

$$= \underset{A}{\operatorname{argmin}} \frac{1}{n} \sum_{i=1}^{n} \left\| y_i - W_{t-1}^\top (\Phi_{t-1}(x_i) + Af_t(x_i)) \right\|^2$$

$$= \underset{A}{\operatorname{argmin}} \frac{1}{n} \sum_{i=1}^{n} \left\| r_i - W_{t-1}^\top Af_t(x_i) \right\|^2, \qquad (5)$$

where $r_i = y_i - W_{t-1}^\top \Phi_{t-1}(x_i)$ are the residuals of the model at layer $t-1$. We term problems of the form (5) **sandwiched least squares problems**, for which Theorem 3.1 below provides the existence of closed-form analytical solutions which are fast to compute.

**Step 2:** After computing $A_t$, we obtain the depth $t$ representation $\Phi_t = \Phi_{t-1} + \eta A_t f_t$. The top-level linear regressor $W_{t+1}$ is then updated via multiple least squares:

$$W_t = \underset{W \in \mathcal{W}}{\operatorname{argmin}} \frac{1}{n} \sum_{i=1}^{n} \left\| y_i - W^\top \Phi_t(x_i) \right\|^2. \qquad (6)$$

In practice, $\ell_2$ regularization is added to equations (5) and (6). Steps 1 and 2 are repeated for $T$ layers. The complete procedure is detailed in Algorithm 1.

**Theorem 3.1.** *Let $r_i \in \mathbb{R}^d, W \in \mathbb{R}^{D \times d}, z_i \in \mathbb{R}^p$ for all $i \in [n]$. Let $\lambda > 0$. Consider the setting of scalar $A$, diagonal $A$, and dense $A \in \mathbb{R}^{D \times p}$. Write $Z \in \mathbb{R}^{N \times p}$ and $R \in \mathbb{R}^{N \times d}$ for the stacked data and residual matrices. Then the minimum of*

$$J(A) = \frac{1}{n} \sum_{i=1}^{n} \left\| r_i - W^\top A z_i \right\|^2 + \lambda \|A\|_F^2$$

*is in the scalar, diagonal, and dense case attained at*

$$A_{\text{scalar}} = \frac{\langle R, ZW \rangle_F}{\|ZW\|_F^2 + n\lambda},$$

$$A_{\text{diag}} = (WW^\top \odot Z^\top Z + \lambda I)^{-1} \operatorname{diag}(WR^\top Z),$$

$$A_{\text{dense}} = U \left[ U^\top WR^\top ZV \oslash \left( \lambda N\mathbf{1} \right. \right.$$
$$\left. \left. + \operatorname{diag}(\Lambda^W) \otimes \operatorname{diag}(\Lambda^Z) \right) \right] V^\top,$$

*where $\| \cdot \|_F$ is the Frobenius norm, $\oslash$ denotes element-wise division, $\otimes$ is the outer product, $\mathbf{1}$ is a matrix of ones, and $WW^\top = U\Lambda^W U^\top$ and $Z^\top Z = V\Lambda^Z V^\top$ are spectral decompositions.*

---

**Algorithm 1** Greedy RFRBoost — MSE Loss

---

**Input:** Data $(x_i, y_i)_{i=1}^n$, $T$ layers, learning rate $\eta$, $\ell_2$ regularization $\lambda$, initial representation $\Phi_0$.
$W_0 \leftarrow \underset{W}{\operatorname{argmin}} \frac{1}{n} \sum_{i=1}^n \|y_i - W^\top \Phi_0(x_i)\|^2$
**for** $t = 1$ **to** $T$ **do**
    Generate random features $f_{t,i} = f_t(x_i, \Phi_{t-1}(x_i))$
    Compute residuals $r_i \leftarrow y_i - W_{t-1}^\top \Phi_{t-1}(x_i)$
    Solve sandwiched least squares
    $A_t \leftarrow \underset{A}{\operatorname{argmin}} \frac{1}{n} \sum_{i=1}^n \|r_i - W_{t-1}^\top A f_{t,i}\|^2 + \lambda\|A\|_F^2$
    Build ResNet layer $\Phi_t \leftarrow \Phi_{t-1} + \eta A_t f_t$
    Update top-level linear regressor
    $W_t \leftarrow \underset{W}{\operatorname{argmin}} \frac{1}{n} \sum_{i=1}^n \|y_i - W^\top \Phi_t(x_i)\|^2 + \lambda\|W\|_F^2$
**end for**
**Output:** ResNet $\Phi_T$ and regressor head $W_T$.

---

**Algorithm 2** Gradient RFRBoost — General Loss

---

**Input:** Data $(x_i, y_i)_{i=1}^n$, loss $l$, $T$ layers, learning rate $\eta$, $\ell_2$ regularization $\lambda$, initial representation $\Phi_0$.
$W_0 \leftarrow \underset{W}{\operatorname{argmin}} \frac{1}{n} \sum_{i=1}^n l(W^\top \Phi_0(x_i), y_i)$
**for** $t = 1$ **to** $T$ **do**
    Generate random features $f_{t,i} = f_t(x_i, \Phi_{t-1}(x_i))$
    Gradient $G_i \leftarrow W_{t-1}\nabla_1 l(W_{t-1}^\top \Phi_{t-1}(x_i), y_i)$
    Fit gradient $A_t \leftarrow least\_squares(f_t, -\frac{\sqrt{n}G}{\|G\|_F}, \lambda)$
    Solve line search with convex solver
    $\alpha_t \leftarrow \underset{\alpha \in \mathbb{R}}{\operatorname{argmin}} \frac{1}{n} \sum_{i=1}^n l(W^\top(\Phi_{t-1} + \alpha A_t f_t), y_i)$
    Build ResNet layer $\Phi_t \leftarrow \Phi_{t-1} + \eta \alpha_t A_t f_t$
    Update top $W_t \leftarrow \underset{W}{\operatorname{argmin}} \frac{1}{n} \sum_{i=1}^n l(W^\top \Phi_t(x_i), y_i)$
**end for**
**Output:** ResNet $\Phi_T$ and top linear layer $W_T$.

---

*Proof.* See Propositions A.1 to A.3 in the Appendix. $\qquad \square$

### 3.3. Gradient Boosting Random Features

While the greedy approach provides optimal solutions for MSE loss, many applications require more general loss functions, such as cross-entropy loss for classification. In such cases, we turn to the gradient-greedy strategy. Recall that in this setting, we aim to minimize the first-order functional Taylor expansion of the risk:

$$\mathcal{R}(W, \Phi + g) \approx \mathcal{R}(W, \Phi) + \langle g, \nabla_2 \mathcal{R}(W, \Phi) \rangle_{L_2^D(\mu)},$$

which holds for functions $g$ with small $L_2^D(\mu)$-norm. As discussed in the context of general gradient representation boosting, a potential issue with directly minimizing $\langle g, \nabla_2 \widehat{\mathcal{R}}(W, \Phi) \rangle_{L_2^D(\widehat{\mu})}$ is that $g$ might learn to maximize its magnitude without following the direction of the functional gradient. To address this, we constrain the problem to ensure that $g$ maintains a unit norm in $L_2^D(\widehat{\mu})$. If we restrict $g$

to residual blocks of the form $g = Af$, where $A \in \mathbb{R}^{D \times p}$ is a linear map, and $f \in \mathbb{R}^p$ are random features, then solving the constrained $L_2^D(\widehat{\mu})$-inner product minimization problem becomes equivalent to solving a quadratically constrained least squares problem. See Appendix B for the proof.

**Theorem 3.2.** *Let $\widehat{\mu} = \frac{1}{n} \sum_{i=1}^n \delta_{(x_i, y_i)}$ be the empirical measure of the data. Then*

$$\underset{A \in \mathbb{R}^{D \times p} \text{ such that } \|Af\|_{L_2^D(\widehat{\mu})} \leq 1}{\operatorname{argmax}} \left\langle Af, \nabla_2 \widehat{\mathcal{R}}(W, \Phi) \right\rangle_{L_2^D(\widehat{\mu})}$$

*is the solution to the quadratically constrained least squares problem*

$$\text{minimize} \quad \frac{1}{n} \sum_{i=1}^n \|\nabla_2 \widehat{\mathcal{R}}(W, \Phi)(x_i) - Af(x_i)\|^2,$$

$$\text{subject to} \quad \frac{1}{n} \sum_{i=1}^n \|Af(x_i)\|^2 = 1,$$

*which in particular has the closed form analytical solution*

$$A = \frac{\sqrt{n}}{\|G\|_{Frobenius}} G^\top F (F^\top F)^{-1}$$

*when $F$ has full rank, where $F \in \mathbb{R}^{n \times p}$ is the feature matrix, and $G \in \mathbb{R}^{n \times D}$ is the matrix given by $G_{i,j} = \left(\nabla_2 \widehat{\mathcal{R}}(W, \Phi)(x_i)\right)_j$.*

The procedure for using Gradient RFRBoost at layer $t$ to build a random feature ResNet $\Phi_t$ is outlined below (see also Figure 1):

**Step 1:** Generate random features $f_t$. Compute the data matrix of the functional gradient $G_{i,j} = \left(\nabla_2 \widehat{\mathcal{R}}(W_{t-1}, \Phi_{t-1})(x_i)\right)_j$. For MSE loss, this is given by

$$G_{i,j}^{\text{MSE}} = \left(W_{t-1}(W_{t-1}^\top \Phi_{t-1}(x_i) - y_i)\right)_j,$$

and for negative cross-entropy loss by

$$G_{i,j}^{\text{CCE}} = \left(W_{t-1}(s(W_{t-1}^\top \Phi_{t-1}(x_i)) - e_{y_i})\right)_j,$$

where $s$ is the softmax function, and $e_{y_i}$ is the one-hot vector for label $y_i$. For full derivation, see Appendix C. We then fit $A_t$ by fitting $f_{t+1}$ to the negative normalized functional gradient $-\frac{\sqrt{n}G}{\|G\|_F}$ via multiple least squares, according to Theorem B.1. In practice, we also include $\ell_2$ regularization.

**Step 2:** Find the optimal *amount of say* $\alpha_t$ via line search

$$\alpha_t = \underset{\alpha > 0}{\operatorname{argmin}} \widehat{\mathcal{R}}(W_{t-1}, \Phi_{t-1} + \alpha A_t f_t)$$

using Theorem 3.1 for MSE loss, or via a suitable convex optimizer such as Newton's method for cross-entropy loss.

**Step 3:** Update the feature representation $\Phi_t = \Phi_{t-1} + \eta \alpha_t A_t f_t$, and the top-level linear predictor

$$W_t = \underset{W \in \mathcal{W}}{\operatorname{argmin}} \widehat{\mathcal{R}}(W, \Phi_t)$$

using an appropriate convex minimizer depending on the specific loss function, such as L-BFGS. The full gradient-greedy procedure is detailed in Algorithm 2.

### 3.4. Theoretical Guarantees

A key advantage of using RFRBoost to construct ResNets over traditional end-to-end trained networks is that RFR-Boost inherits strong theoretical guarantees from boosting theory. We analyze RFRBoost within the theoretical framework of Generalized Boosting (Suggala et al., 2020), where excess risk bounds have been established in terms of modular Rademacher complexities of the family of weak learners $\mathcal{G}_t$ at each boosting iteration $t$. These bounds are based on the $(\beta, \epsilon)$-weak learning condition, defined below.

**Definition 3.3** (Suggala et al. (2020)). *Let $\beta \in (0, 1]$ and $\epsilon \geq 0$. We say that $\mathcal{G}_{t+1}$ satisfies the $(\beta, \epsilon)$-weak learning condition if there exists a $g \in \mathcal{G}_{t+1}$ such that*

$$\frac{\left\langle g, -\nabla_2 \mathcal{R}(W_t, \Phi_t) \right\rangle_{L_2^D(\mu)}}{\beta \sup_{g \in \mathcal{G}_{t+1}} \|g\|_{L_2^D(\mu)}} + \epsilon \geq \left\|\nabla_2 \mathcal{R}(W_t, \Phi_t)\right\|_{L_2^D(\mu)}.$$

Intuitively, this condition states that there exists a weak feature transformation in $\mathcal{G}_{t+1}$ that is negatively correlated with the functional gradient. In the sample-splitting setting, where each boosting iteration uses an independent sample of size $\widetilde{n} = \lfloor n/T \rfloor$, we derive the following regret bound for RFRBoost. This bound describes how the excess risk decays compared to an optimal predictor as $T$ and $\widetilde{n}$ vary.

**Theorem 3.4.** *Let $l$ be $L$-Lipschitz and $M$-smooth with respect to the first argument. For a matrix $W$, let $\lambda_{\min}(W)$ and $\lambda_{\max}$ denote its smallest and largest singular values, respectively. Consider RFRBoost with the hypothesis set of linear predictors $\mathcal{W} = \left\{W \in \mathbb{R}^{D \times d} : \lambda_{\min}(W) > \lambda_0 > 0, \lambda_{\max}(W) < \lambda_1\right\}$, and weak feature transformations $\mathcal{G}_t = \{x \mapsto A \tanh(B\Phi_{t-1}(x)) : \lambda_{\max}(A), \lambda_{\max}(B) < \lambda_1\}$ satisfying the $(\beta, \epsilon_t)$-weak learning condition. Let the boosting learning rates be $\eta_t = ct^{-s}$ for some $s \in \left(\frac{\beta+1}{\beta+2}, 1\right)$ and $c > 0$. Then $T$-layer RFRBoost satisfies the following risk bound for any $W^*, \Phi^*$, and $a \in \left(0, \beta(1-s)\right)$ with probability at least $1 - \delta$ over datasets of size $n$:*

$$\mathcal{R}(W_T, \Phi_T) \leq \mathcal{R}(W^*, \Phi^*) + 2\sum_{t=1}^T \eta_t \epsilon_t$$

$$+ C\left(\frac{1}{T^a} + \frac{T^{2-s}\left(1 + \sqrt{\log \frac{T}{\delta}}\right)}{\sqrt{\widetilde{n}}}\right),$$

*where the constant $C$ does not depend on $T$, $n$, and $\delta$.*

*Proof.* See Appendix D. □

For similar results in the literature, see for instance Suggala et al. (2020), Corollary 4.3.

### 3.5. Time Complexity

The serial time complexity of RFRBoost with MSE loss, assuming tabular data and dense random features, is $O(T[N(D^2 + Dd + p^2 + pD) + D^3 + dD^2 + p^3 + Dp^2])$, as derived from Algorithm 2. Here $N$ is the dataset size, $D$ is the dimension of the neural network representation, $d$ is the output dimension of the regression task, $p$ is the number of random features, and $T$ is the number of layers (boosting rounds). The computation is dominated by matrix operations, which are well-suited for GPU acceleration. The classification case follows similarly.

## 4. Numerical Experiments

In this section, we compare RFRBoost against a set of baseline models on a wide range of tabular regression and classification datasets, as well as on a challenging synthetic point cloud separation task. All our code is publicly available at https://github.com/nikitazozoulenko/random-feature-representation-boosting.

### 4.1. Tabular Regression and Classification

**Datasets:** We experiment on all datasets of the curated OpenML tabular regression (Fischer et al., 2023) and classification (Bischl et al., 2021) benchmark suites with 200 or fewer features. This amounts to a total of 91 datasets per model. Due to the large number of datasets, we limit ourselves to 5000 observations per dataset. We preprocess each dataset by one-hot encoding all categorical variables and normalizing all numerical features to have zero mean and variance one.

**Evaluation Procedure:** We use a nested 5-fold cross-validation (CV) procedure to tune and evaluate all models, run independently for each dataset. The innermost CV is used to tune all hyperparameters, for which we use the Bayesian hyperparameter tuning library Optuna (Akiba et al., 2019). For regression, we use MSE loss, and for classification, we use cross-entropy loss. All experiments are run on a single CPU core on an institutional HPC cluster, mostly comprised of AMD EPYC 7742 nodes. We report the average test scores for each model, as well as mean training times for a single fit. The average relative rank of each model is presented in a critical difference diagram, indicating statistically significant clusters based on a Wilcoxon signed-rank test with Holm correction. This

is a well-established methodology for comparing multiple algorithms across multiple datasets (Demšar, 2006; García & Herrera, 2008; Benavoli et al., 2016).

**Baseline Models:** We compare RFRBoost, a random feature ResNet, against several strong baselines, including end-to-end (E2E) trained MLP ResNets, single-layer random feature neural networks (RFNNs), ridge regression, logistic regression, and XGBoost (Chen & Guestrin, 2016). The E2E ResNets are trained using the Adam optimizer (Kingma & Ba, 2015) with cosine learning rate annealing, ReLU activations, and batch normalization. While XGBoost is a powerful gradient boosting model, it differs fundamentally from RFRBoost. XGBoost ensembles a large number of weak decision trees (up to 1000 in our experiments) to build a strong predictor. RFRBoost, on the other hand, uses gradient representation boosting to construct a small number of residual blocks, followed by a single linear predictor. For RFRBoost and RFNNs we use SWIM random features with tanh activations, as detailed in Appendix E. In the regression setting, we evaluate three variants of RFRBoost: using a scalar, a diagonal, or a dense $A$ matrix. When using a dense $A$, we set the initial mapping $\Phi_0(x)$ to the identity; otherwise, $\Phi_0(x)$ is a randomly initialized dense layer. For classification, we use the gradient-greedy variant of RFRBoost with LBFG-S as the convex solver. We use the official implementation of XGBoost, and implement all other baseline models in PyTorch (Paszke et al., 2019).

**Hyperparameters:** All hyperparameters are tuned with Optuna in the innermost fold, using 100 trials per outer fold, model, and dataset. For ridge and logistic regression, we tune the $\ell_2$ regularization. For the neural network-based models, we fix the feature dimension of each residual block to 512 and use 1 to 10 layers. For E2E networks, we tune the hidden size, learning rate, learning rate decay, number of epochs, batch size, and weight decay. For RFRBoost, we tune the $\ell_2$ regularization of the linear predictor and functional gradient mapping, the boosting learning rate, and the variance of the random features. For RFNNs, we tune the random feature dimension, random feature variance, and $\ell_2$ regularization. For XGBoost, we tune the $\ell_1$ and $\ell_2$ regularization, tree depth, boosting learning rate, and the number of weak learners. For a detailed list of hyperparameter ranges, along with an ablation study comparing SWIM random features to i.i.d. Gaussian random features, we refer the reader to Appendix E.

**Results:** Summary results for regression and classification are presented in Tables 1 to 2 and Figures 2 to 3, respectively. Full dataset-wise results are reported in Appendix E. We find that RFRBoost ranks higher than all other baseline models. For regression tasks, the gradient-greedy version of RFRBoost outperforms the exact-greedy variant, contrary to the observations of Suggala et al. (2020) for SGD-trained gradient representation boosting. This difference likely arises

*Table 1.* Average test RMSE and single-core CPU fit times on the OpenML regression datasets.

| MODEL | MEAN RMSE | FIT TIME (S) |
|---|---|---|
| GRADIENT RFRBOOST | 0.408 | 1.688 |
| GREEDY RFRBOOST $A_{dense}$ | 0.408 | 2.734 |
| GREEDY RFRBOOST $A_{diag}$ | 0.415 | 1.631 |
| GREEDY RFRBOOST $A_{scalar}$ | 0.434 | 1.024 |
| XGBOOST | 0.394 | 1.958 |
| E2E MLP RESNET | 0.412 | 19.309 |
| RFNN | 0.434 | 0.053 |
| RIDGE REGRESSION | 0.540 | 0.001 |

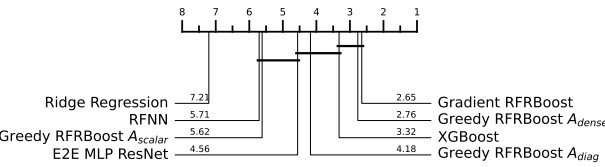

*Figure 2.* Critical difference diagram based on pairwise relative rank of test RMSE. Bars indicate no significant difference ($\alpha = 0.05$). The average rank is displayed for each model.

because the SGD-based approach does not incorporate the $L_2^D(\mu)$-norm constraint during training, which is crucial for preserving the functional direction of the residual block. The test scores follow the ordering $A_{\text{dense}} > A_{\text{diag}} > A_{\text{scalar}}$, demonstrating that RFRBoost is more expressive when mapping random feature layers to the functional gradient, rather than simply stacking random feature layers. While XGBoost achieves a slightly lower RMSE than RFRBoost, it performs worse in terms of average rank. Moreover, RFRBoost significantly outperforms both RFNNs and E2E MLP ResNets, while being an order of magnitude faster to train than the latter. Although the reported training times are CPU-based, our implementation suggests both methods would benefit similarly from GPU acceleration, making the presented times representative.

### 4.2. Point Cloud Separation

We evaluate RFRBoost on a challenging synthetic dataset originating from the neural ODE literature (Sander et al., 2021). The dataset consists of 10,000 points sampled from concentric circles, and the task is to linearly separate (i.e. classify) the concentric circles while restricting the ResNet hidden size to 2, see Figure 4. We compare RFRBoost to E2E MLP ResNets, while also presenting classification results for logistic regression and RFNNs in Table 3. All models were trained with cross entropy loss, using a 5-fold CV grid search for hyperparameter tuning. For E2E

*Table 2.* Average test accuracies and single-core CPU fit times on the OpenML classification datasets.

| MODEL | MEAN ACC | FIT TIME (S) |
|---|---|---|
| RFRBOOST | 0.853 | 2.519 |
| XGBOOST | 0.853 | 3.859 |
| E2E MLP RESNET | 0.851 | 20.881 |
| RFNN | 0.845 | 1.189 |
| LOGISTIC REGRESSION | 0.821 | 0.165 |

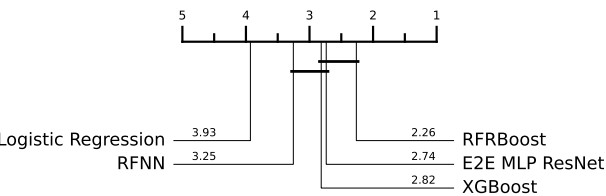

*Figure 3.* Critical difference diagram based on pairwise relative rank of test accuracy. Bars indicate no significant difference ($\alpha = 0.05$). The average rank is displayed for each model.

*Table 3.* Average test accuracies on the concentric circles point cloud separation task, averaged across 10 runs.

| MODEL | MEAN ACC | STD DEV |
|---|---|---|
| RFRBOOST | 0.997 | 0.002 |
| RFNN | 0.887 | 0.037 |
| E2E MLP RESNET | 0.732 | 0.144 |
| LOGISTIC REGRESSION | 0.334 | 0.023 |

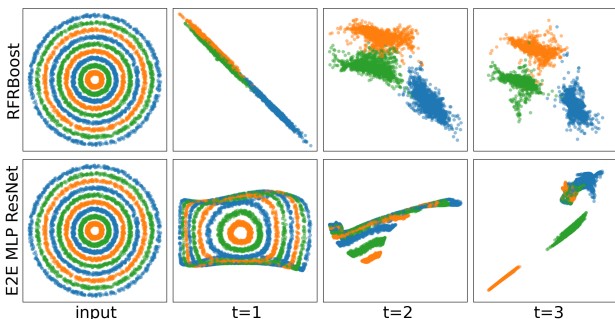

*Figure 4.* Point cloud separation of test data at each layer.

MLP ResNets, the learning rate was tuned, while for the other models, only the $\ell_2$ regularization of the classification head was tuned. The hidden size, residual block feature dimension, and activation function was fixed at 2, 512, and $\tanh$, respectively, for all models. See Appendix E.3 for more details.

Notably, the RFNN, which uses SWIM random features to map the initial 2-dimensional input to a higher-dimensional space before applying a linear classifier, fails to classify all points correctly. RFRBoost, in contrast, achieves near-perfect linear separation by first mapping the random features to the functional gradient of the network representation, before applying a 2-dimensional linear classifier. This result is somewhat surprising because RFRBoost does not rely on explicit Fourier features or a learnt radial basis, demonstrating the power of gradient representation boosting. It further illustrates how RFRBoost can solve problems that are intractable for E2E-trained networks and RFNNs. Figure 4 shows that the E2E MLP ResNet correctly separates only two of the nine concentric rings, similar to the failure to converge observed by Sander et al. (2021).

### 4.3. Experiments on Larger-scale Datasets

To complement our experiments on the OpenML benchmark suite, we conducted additional full-scale evaluations on four larger datasets (two with $\sim$100k samples, two with $\sim$500k samples) to assess performance and scalability in larger data regimes. Full experimental details, including dataset splits, hyperparameter grids, and plots of training time and predictive performance versus training set size, are provided in Appendix E.5. We find that RFRBoost significantly outperforms traditional single-layer RFNNs across all tested datasets and training sizes, particularly as dataset size increases, highlighting its effectiveness in leveraging depth for random feature models. While RFRBoost performs strongly against E2E trained networks in medium-sized data regimes, our findings indicate that E2E networks and XGBoost eventually outperform RFRBoost as dataset size increases on 3 out of 4, and 2 out of 4 datasets, respectively. We hypothesize that using RFRBoost as an initialization strategy, followed by end-to-end fine-tuning, could be a promising direction to further enhance the performance of deep networks. We leave this to future work.

## 5. Conclusion

This paper introduced RFRBoost, a novel method for constructing deep residual random feature neural networks (RFNNs) using boosting theory. RFRBoost addresses the limitations of single-layer RFNNs by using random features to learn optimal ResNet-like residual blocks that approximate the negative functional gradient at each layer of the network, thereby enhancing performance while retaining the computational benefits of convex optimization for RFNNs. This procedure can be viewed as performing functional gradient descent on the network neurons, which has connections to gradient boosting theory, as opposed to classical SGD which performs gradient descent on the network weights and biases. In our framework, we derived closed-form solutions for greedy layer-wise boosting with MSE loss, and presented a general fitting algorithm for arbitrary loss functions based on solving a quadratically constrained least squares problem. Through extensive numerical experiments on tabular datasets for both regression and classification, we demonstrated that RFRBoost significantly outperforms traditional RFNNs and end-to-end trained MLP ResNets in the small- to medium-scale regime where RFNNs are typically applied, while offering substantial computational advantages, and theoretical guarantees stemming from boosting theory. RFRBoost represents a significant step towards building powerful, stable, efficient, and theoretically sound deep networks using untrained random features. Future work will focus on extending RFRBoost to other domains such as time series or image data, exploring different types of random features and momentum strategies, implementing more efficient GPU acceleration, scaling to large datasets, and using RFRBoost as an initialization strategy for large-scale end-to-end training.

## Acknowledgements

TC has been supported by the EPSRC Programme Grant EP/S026347/1. NZ has been supported by the Roth Scholarship at Imperial College London, and acknowledges conference travel support from G-Research. We acknowledge computational resources and support provided by the Imperial College Research Computing Service (DOI: `10.14469/hpc/2232`). For the purpose of open access, the authors have applied a Creative Commons Attribution (CC BY) licence to any Author Accepted Manuscript version arising.

## Impact Statement

This paper presents work whose goal is to advance the field of Machine Learning. There are many potential societal consequences of our work, none which we feel must be specifically highlighted here.

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

# A. Analytic Solutions to Sandwiched Least Squares Problems

In this section, we derive the analytic closed form expressions of the sandwiched least squares problems presented in Theorem 3.1. We use NumPy notation for element and row indexing, i.e. if $X$ is a matrix, then $X_i$ denotes the $i$'th row of $X$. $\|\cdot\|_F$ denotes the Frobenius norm.

**Proposition A.1 (Scalar case).** *Let $R \in \mathbb{R}^{n \times d}, W \in \mathbb{R}^{D \times d}, A \in \mathbb{R}$, and $X \in \mathbb{R}^{n \times D}$. Let $\lambda > 0$. Then the minimum of*

$$J(A) = \frac{1}{n} \sum_{i=1}^{n} \left\| R_i - W^\top A X_i \right\|^2 + \lambda A^2$$

*is uniquely attained at*

$$A_{\text{scalar}} = \frac{\langle R, XW \rangle_F}{\|XW\|_F^2 + n\lambda} = \frac{\frac{1}{n} \sum_{i=1}^{n} \langle W^\top X_i, R_i \rangle}{\frac{1}{n} \sum_{i=1}^{n} \|W^\top X_i\|^2 + \lambda}.$$

*Proof.* We first rewrite the objective $J(A)$ using the Frobenius norm

$$J(A) = \frac{1}{n} \|XAW - R\|_F^2 + \lambda A^2.$$

Differentiating with respect to $A \in \mathbb{R}$ and setting $J'(A) = 0$ gives

$$0 = J'(A) = \frac{2}{n}(XW)^\top(AXW - R) + 2\lambda A$$

$$\Longleftrightarrow \quad A\|XW\|_F^2 + n\lambda A = \langle XW, R \rangle_F,$$

from which the result follows by factoring out $A$. Note that the second derivative is

$$J''(A) = \frac{2}{n}\|XW\|_F^2 + 2\lambda > 0,$$

hence the problem is convex and the local minima is the unique global minima. □

Python NumPy code for this is

```
XW = X @ W
top = np.sum(R * XW) / n
bot = np.sum(XW * XW) / n
A = top / (bot + l2_reg)
```

**Proposition A.2 (Diagonal case).** *Let $R \in \mathbb{R}^{n \times d}, W \in \mathbb{R}^{D \times d}, A = \text{diag}(a_1, ..., a_D) \in \mathbb{R}^{D \times D}$, and $X \in \mathbb{R}^{n \times D}$. Let $\lambda > 0$. Then the minimum of*

$$J(A) = \frac{1}{n} \sum_{i=1}^{n} \left\| R_i - W^\top A X_i \right\|^2 + \lambda \|A\|_F^2$$

*is uniquely attained by the solution to the system of linear equations*

$$b = (C + \lambda I)A,$$

*where*

$$C = WW^\top \odot X^\top X, \qquad b = \text{diag}(WR^\top X).$$

*Proof.* We expand $J(A)$ and find that

$$J(A) = J(a_1, ..., a_D)$$
$$= \frac{1}{n} \sum_{i=1}^{n} \left\| R_i - W^\top A X_i \right\|^2 + \lambda \|A\|_F^2$$
$$= \frac{1}{n} \sum_{i=1}^{n} \left( R_i - W^\top [a_1 X_{i,1}, ..., a_D X_{i,D}]^\top \right)^\top \left( R_i - W^\top [a_1 X_{i,1}, ..., a_D X_{i,D}]^\top \right) + \lambda \sum_{k=1}^{D} a_k^2.$$

Differentiating with respect to a specific $a_k$ gives

$$0 = \frac{\partial J(a_1, ..., a_D)}{\partial a_k} = \frac{1}{n} \sum_{i=1}^{n} -2 \left( W_k X_{i,k} \right)^\top \left( R_i - W^\top [a_1 X_{i,1}, ..., a_D X_{i,D}]^\top \right) + 2\lambda a_k$$

$$\iff \sum_{i=1}^{n} (W_k X_{i,k})^\top R_i = \sum_{i=1}^{n} \left[ (W_k X_{i,k})^\top W^\top A X_i + \lambda a_k \right]$$
$$= \sum_{i=1}^{n} \left[ \lambda a_k + \sum_{j=1}^{D} (W_k X_{i,k})^\top W_j X_{i,j} a_j \right].$$

This implies that the value of $A$ which minimizes the objective is the solution to the system of linear equations given by

$$b = (C + \lambda I)A,$$

where for $k, j \in [D]$

$$C_{k,j} = \sum_{i=1}^{n} \left( W_k X_{i,k} \right)^\top W_j X_{i,j}, \qquad b_k = \sum_{i=1}^{n} R_i^\top W_k X_{i,k}.$$

Simplifying we obtain that

$$C = WW^\top \odot X^\top X, \qquad b = \text{diag}(WR^\top X).$$

This solution is unique since the objective is strictly convex with hessian given by $\frac{2}{n}C + 2\lambda I$. $\qquad \square$

Python code for this is

```
b = np.mean( (R @ W.T) * X, axis=0)
C = (W @ W.T) * (X.T @ X) / n
A = np.linalg.solve(C + l2_reg * np.eye(D), b)
```

**Proposition A.3 (Dense case).** *Let $R \in \mathbb{R}^{n \times d}, W \in \mathbb{R}^{D \times d}, A \in \mathbb{R}^{D \times p}$, and $X \in \mathbb{R}^{n \times D}$. Let $\lambda > 0$. Then the minimum of*

$$J(A) = \frac{1}{n} \sum_{i=1}^{n} \left\| r_i - W^\top A x_i \right\|^2 + \sum_{k=1}^{D} \sum_{j=1}^{p} \lambda A_{k,j}^2$$
$$= \frac{1}{n} \|W^\top A X^\top - R^\top\|_F^2 + \lambda \|A\|_F^2$$

*is uniquely obtained by solving the system of linear equations given by*

$$WR^\top X = WW^\top A X^\top X + \lambda n A,$$

*which can be solved using the spectral decompositions $WW^\top = U\Lambda^W U^\top$ and $X^\top X = V\Lambda^X V^\top$,*

$$A_{\text{dense}} = U \left[ U^\top W R^\top X V \oslash \left( \lambda n \mathbf{1} + \text{diag}(\Lambda^W) \otimes \text{diag}(\Lambda^X) \right) \right] V^\top,$$

*where $\oslash$ denotes element-wise division, $\otimes$ is the outer product, and $\mathbf{1}$ is a matrix of ones.*

*Proof.* Using the gradient chain rule we find that

$$0 = \nabla J(A)$$
$$= \frac{-2}{n} W(W^\top A X^\top - R^\top) X + 2\lambda A$$
$$\iff W R^\top X = W W A X^\top X + \lambda N A.$$

Letting $WW^\top = U\Lambda^W U^\top$ and $X^\top X = V\Lambda^X V^\top$ be spectral decompositions, and setting $\widetilde{A} = U^\top A V$ we see that

$$U^\top W R^\top X V = \Lambda^W \widetilde{A}\Lambda^X + \lambda n \widetilde{A}.$$

By inspecting this equation element-wise we find that

$$(U^\top W R^\top X V)_{k,j} = \Lambda^W_{k,k}\widetilde{A}_{k,j}\Lambda^X_{j,j} + \lambda n \widetilde{A}_{k,j},$$

which implies that

$$\widetilde{A}_{k,j} = \frac{(U^\top W R^\top X V)_{k,j}}{\Lambda^W_{k,k}\Lambda^X_{j,j} + N\lambda},$$

from which the conclusion follows after change of basis $A = U\widetilde{A}V^\top$. The solution is unique since the problem is strictly convex, which is easily proved using the convexity of the Frobenius norm and the linearity of the matrix expressions involving $A$, together with the strict convexity of the regularization term. $\square$

Python code for this is

```
SW, U = np.linalg.eigh(W @ W.T)
SX, V = np.linalg.eigh(X.T @ X)
A = (U.T @ W @ R.T @ X @ V)
A = A / (n*lambda_reg + SW[:, None]*SX[None, :])
A = U @ A @ V.T
```

## B. Functional Gradient Inner Product

In this section we prove Theorem 3.1, showing that minimizing the $L_2^D(\mu)$ inner product under a norm constraint for a simple random feature residual block is equivalent to solving a quadratically constrained least squares problem. We use the same matrix notation as in Appendix A.

**Theorem B.1.** *Let $\mu = \frac{1}{n}\sum_{i=1}^n \delta_{x_i}$ be an empirical measure, $h \in L_2^D(\mu)$, and $f \in L_2^p(\mu)$. Then solving*

$$\operatorname*{argmin}_{A\in\mathbb{R}^{D\times p} \text{ such that } \|Af\|_{L_2^D(\mu_n)}\leq 1} \langle h, Af\rangle_{L^2(\mu_n)}$$

*is equivalent to solving the quadratically constrained least squares problem*

$$\frac{1}{n}\sum_{i=1}^n \|h(x_i) - Af(x_i)\|^2, \qquad \text{subject to} \qquad \frac{1}{n}\sum_{i=1}^n \|Af(x_i)\|^2 = 1.$$

*In particular, when $F$ is of full rank, we obtain the closed form solution*

$$A = -\frac{\sqrt{n}}{\|H\|_F} H^\top F(F^\top F)^{-1},$$

*where $F \in \mathbb{R}^{n\times p}$ and $H \in \mathbb{R}^{n\times D}$ are the matrices given by $F_{i,j} = f(x_i)_j$ and $H_{i,k} = h(x_i)_k$.*

*Proof.* Using the definition of the $L_2^D(\mu)$ norm for empirical measures, we find that

$$\langle h, Af \rangle_{L_2^D(\mu)} = \frac{1}{n} \sum_{i=1}^{n} \langle h(x_i), Af(x_i) \rangle = \frac{1}{n} \sum_{i=1}^{n} \langle H_i, AF_i \rangle. \tag{7}$$

Furthermore, the constraint can be expressed as

$$\|Af\|_{L_2^D(\mu_n)}^2 = \frac{1}{n} \sum_{i=1}^{n} \|Af(x_i)\|^2 = \frac{1}{n} \sum_{i=1}^{n} \|AF_i\|^2 = 1, \tag{8}$$

with equality instead of inequality, since we always obtain a bigger inner product in magnitude by normalizing by $\|Af\|_{L_2^D(\mu_n)}^2$. Minimizing (7) subject to (8) is equivalent to solving a constrained least squares problem, since we can write

$$\frac{1}{n} \sum_{i=1}^{n} \|H_i - AF_i\|^2 = \frac{1}{n} \sum_{i=1}^{n} \|H_i\|^2 - 2\langle H_i, AF_i \rangle + \|AF_i\|^2,$$

where we see that the first term $\|H_i\|^2$ is constant w.r.t $A$, and the third term gives the constraint. Hence, minimizing the constrained least squares problem is equivalent to maximizing the inner product, and the solution to the original problem is obtained by multiplying the least squares solution by $-1$ since we are interested in the argmin rather than the argmax.

Continuing, to solve the quadratically constrained least squares problem, we introduce the Lagrangian

$$J(A, \nu) = \frac{1}{n} \sum_{i=1}^{n} |H_i - AF_i|^2 - \nu \left( \frac{1}{n} \sum_{i=1}^{n} |AF_i|^2 - 1 \right)$$

$$= \frac{1}{n} \|H^\top - AF^\top\|_F^2 - \nu \left( \frac{1}{n} \|AF^\top\|_F^2 - 1 \right).$$

Differentiating with respect to $A$ gives

$$0 = \nabla_1 J(A, \nu) = \frac{-2}{n}(H^\top - AF^\top)F - \nu \frac{2}{n} AF^\top F,$$

which implies that

$$H^\top F = (1 - \nu)AF^\top F.$$

and

$$A = \frac{1}{1 - \nu} H^\top F (F^\top F)^{-1}$$

$$= \frac{1}{1 - \nu} H^\top U \Lambda^{-1} V^\top,$$

assuming that $\nu \neq 1$ and that $F$ is of full rank, with SVD decomposition $F = U\Lambda V^\top$. The constraint becomes

$$n = \|AF^\top\|_F^2$$

$$= \frac{1}{(1 - \nu)^2} \|H^\top U \Lambda^{-1} V^\top V \Lambda U^\top\|_F^2$$

$$= \frac{1}{(1 - \nu)^2} \|H\|_F^2,$$

therefore $1 - \nu = \pm \frac{\|H\|_F}{\sqrt{n}}$. The solution to the constraint least squares problem is obtained by using the positive sign, hence the solution to the original problem is given by the negative sign.

If $\nu = 1$, then $H^\top F = 0$, implying that $\langle H_i, AF_i \rangle_{L_2^D(\mu)} = 0$ for all matrices $A$. Hence the same closed-form solution holds in this case too. $\qquad \square$

*Remark* B.2. It is clear from the proof how to augment the expression for $A$ when $F$ is not of full rank. However, in practice we instead use ridge regression for increased numerical stability. A similar result as the above can be proven for ridge regression, albeit with a more complicated expression for $A$ involving non-trivial combinations of $\Lambda$ and $\lambda$. We omit this detail here, and simply use ridge regression in practice. Note also that $F(F^\top F)^{-1}$ can be expressed as a pseudo-inverse of $F$, after suitable transpositions of the matrices involved.

# C. Gradient Calculations

For completeness, we derive the functional gradient used in GradientRFRBoost, for MSE loss, categorical cross-entropy loss, and binary cross-entropy loss. Recall that the functional gradient, in all cases, is given by

$$\nabla_2 \mathcal{R}(W, \Phi)(x) = \mathbb{E}_{\mu_{Y|X=x}}[W \nabla_1 L(W^\top \Phi(x), Y)],$$

where $\nabla_i$ denotes the gradient with respect to the $i$'th argument.

## C.1. Mean Squared Error Loss

For regression, we use mean squared error loss $l(x, y) = \frac{1}{2}\|x - y\|^2$. In this case, we find that

$$\nabla_1 l(x, y) = x - y,$$

hence

$$\begin{aligned}
\nabla_2 \mathcal{R}(W, \Phi)(x) &= \mathbb{E}_{Y|X=x}[W \nabla_1 l(W^\top \Phi(x), Y)] \\
&= \mathbb{E}_{Y|X=x}[W(W^\top \Phi(x) - Y)].
\end{aligned}$$

When $\mu = \sum_{i=1}^n \delta_{(x_i, y_i)}$ is an empirical measure, the above reads in matrix form as

$$G = WW^\top X^\top - WY^\top$$

where $X \in \mathbb{R}^{n \times D}$ and $Y \in \mathbb{R}^{n \times d}$ are the matrices given by $X_{i,j} = \Phi_j(x_i)$ and $Y_{i,k} = (y_i)_k$.

## C.2. Binary Cross-Entropy Loss

Denote the binary cross-entropy (BCE) loss as $l(x, y) = -y \log(\sigma(x)) - (1 - y) \log(1 - \sigma(x))$, where $\sigma(x) = \frac{1}{1+e^{-x}}$ is the sigmoid function. The gradient of the BCE loss with respect to the logit $\sigma(x)$ is

$$\frac{\partial l(x, y)}{\partial \sigma(x)} = -\frac{y}{\sigma(x)} + \frac{1 - y}{1 - \sigma(x)} = \frac{\sigma(x) - y}{\sigma(x)(1 - \sigma(x))}.$$

Using the chain rule together with the fact that $\sigma'(x) = \sigma(x)(1 - \sigma(x))$ gives that

$$\nabla_1 l(x, y) = \frac{\partial l(x, y)}{\partial \sigma(x)} \frac{\partial \sigma(x)}{\partial x} = \sigma(p(x)) - y,$$

whence we obtain

$$\nabla_2 \mathcal{R}(W, \Phi)(x) = \mathbb{E}_{\mu_{Y|X=x}} \left[ W(\sigma(W^\top \Phi(x)) - y) \right].$$

## C.3. Categorical Cross-Entropy Loss

The analysis for the multi-class case is similar to the binary case. The cross-entropy loss $l : \mathbb{R}^K \times \{1, \ldots, K\} \to [0, \infty)$ for logits $x$ with true label $y$ is given by

$$l(x, y) = -\log(s_y(x))$$

where $s$ is the softmax function defined by

$$s_y(x) = \frac{\exp(x_y)}{\sum_{j=1}^K \exp(x_j)}.$$

We aim to prove that $\nabla_1 l(x, y) = p(x) - e_y$, where $e_y \in \mathbb{R}^K$ is the one-hot vector for $y$. To see this, consider for any $1 \leq k \leq K$ the following:

$$
\begin{aligned}
\frac{\partial l(x, y)}{\partial x_k} &= -\frac{\partial}{\partial x_k} \log(s_y(x)) \\
&= -\frac{1}{s_y(x)} \frac{\partial}{\partial x_k} s_y(x) \\
&= -\frac{1}{s_y(x)} \frac{\partial}{\partial x_k} \frac{\exp(x_y)}{\sum_{j=1}^K \exp(x_j)} \\
&= -\frac{1}{s_y(x)} \left( \frac{\mathbf{1}_{y=k} \exp(x_y) \sum_{j=1}^K \exp(x_j) - \exp(x_y) \exp(x_k)}{\left( \sum_{j=1}^K \exp(x_j) \right)^2} \right) \\
&= -\frac{1}{s_y(x)} \left( \mathbf{1}_{y=k} s_y(x) - s_y(x) s_k(x) \right) \\
&= s_k(x) - \mathbf{1}_{y=k}
\end{aligned}
$$

Since this holds for all $k$, we have that

$$
\nabla_1 l(x, y) = s(x) - e_y.
$$

Then,

$$
\begin{aligned}
\nabla_2 \mathcal{R}(W, \Phi)(x) &= \mathbb{E}_{\mu_{Y|X=x}}[W \nabla_1 l(W^\top \Phi(x), Y)] \\
&= \mathbb{E}_{\mu_{Y|X=x}}[W(s(W^\top \Phi(x)) - e_Y)].
\end{aligned}
$$

When $\mu = \sum_{i=1}^n \delta_{(x_i, y_i)}$ is an empirical measure, the above reads in matrix form as

$$
G = W(P(X) - E_Y)^\top,
$$

where $P(X) \in \mathbb{R}^{n \times K}$ is the matrix given by $P(X)_{i,k} = s_k(W^\top \Phi(x_i))$, and $E_Y \in \mathbb{R}^{n \times K}$ is the matrix given by $(E_Y)_{i,k} = \mathbf{1}_{y_i=k}$.

## D. Excess Risk Bound

In this section we study the excess risk bound of RFRBoost in the framework of Generalized Boosting (Suggala et al., 2020). We take a slightly different approach in our proofs which streamlines the process: instead of bounding the $\| \cdot \|_1$ norm of the rows of the weight matrices, we instead bound the maximum singular values $\sigma_{\max}$.

### D.1. Preliminaries

We repeat the following definition from Section 3.

**Definition D.1** (Suggala et al. (2020))**.** Let $\beta \in (0, 1]$ and $\epsilon \geq 0$. We say that $\mathcal{G}_{t+1}$ satisfies the $(\beta, \epsilon)$-weak learning condition if there exists a $g \in \mathcal{G}_{t+1}$ such that

$$
\frac{\langle g, -\nabla_2 \mathcal{R}(W_t, \Phi_t) \rangle_{L_2^D(\mu)}}{\beta \sup_{g \in \mathcal{G}_{t+1}} \|g\|_{L_2^D(\mu)}} + \epsilon \geq \|\nabla_2 \mathcal{R}(W_t, \Phi_t)\|_{L_2^D(\mu)}.
$$

Consider the sample-splitting variant of boosting, where at each boosting iteration we use an independent sample of size $\widetilde{n} = n/T$. Let $\mu_t = \sum_{i=1}^{\widetilde{n}} \delta_{(x_{t,i}, y_{t,i})}$ denote the empirical measure of the $t$-th independent sample. The risk bounds in the sequel depend on Rademacher complexities related to the class of weak feature transformations $\mathcal{G}_t$ and set of linear predictors $\mathcal{W}$, which we define below:

$$
\mathfrak{R}(\mathcal{W}, \mathcal{G}_t) = \mathbb{E}_\rho \left[ \sup_{\substack{W \in \mathcal{W} \\ g \in \mathcal{G}_t}} \frac{1}{\widetilde{n}} \sum_{i=1}^{\widetilde{n}} \sum_{k=1}^d \rho_{i,k} [W^\top g(x_{t,i})]_k \right],
$$

$$\mathfrak{R}(\mathcal{G}_t) = \mathbb{E}_\rho \left[ \sup_{g \in \mathcal{G}_t} \frac{1}{\widetilde{n}} \sum_{i=1}^{\widetilde{n}} \sum_{j=1}^{D} \rho_{i,j} [g(x_{t,i})]_k \right],$$

where $\rho_{i,j}$ are Rademacher random variables, that is, independent random variables taking values 1 and $-1$ with equal probability.

The excess risk bound of RFRBoost is based on the following result:

**Theorem D.2** (Suggala et al. (2020)). *Suppose that the loss $l$ is $L$-Lipschitz and $M$-smooth with respect to the first argument. Let the hypothesis set of linear predictors $\mathcal{W}$ be such that all $W \in \mathcal{W}$ satisfy $\lambda_{\min}(W^\top W) \geq \sigma_{\min}^2 > 0$ and $\lambda_{\max}(W^\top W) \leq \sigma_{\max}^2$. Moreover, suppose for all $t$ that $\mathcal{G}_t$ satisfies the $(\beta, \epsilon_t)$-weak learning condition for $\mu_t$, and that all $g \in \mathcal{G}_t$ are bounded with $\sup_X \|g(X)\|_2 \leq R$. Let the boosting learning rates $(\eta_t)_{t=1}^T$ be $\eta_t = ct^{-s}$ for some $s \in \left(\frac{\beta+1}{\beta+2}, 1\right)$ and $c > 0$. Then both the exact-greedy and gradient-greedy representation boosting algorithms of Section 2.2 satisfy the following risk bound for any $W^*, \Phi^*$, and $a \in \left(0, \beta(1-s)\right)$, with probability at least $1 - \delta$ over datasets of size $n$:*

$$\mathcal{R}(W_T, \Phi_T) \leq \mathcal{R}(W^*, \Phi^*) + \mathcal{O}\left(\frac{1}{T^a} + T^{2-s}\sqrt{\frac{\log \frac{T}{\delta}}{\widetilde{n}}}\right) + 2\sum_{t=1}^T \eta_t \left(L\mathfrak{R}(\mathcal{W}, \mathcal{G}_t) + L\mathfrak{R}(\mathcal{G}_t) + \epsilon_t\right).$$

We aim to apply the excess risk bound of Theorem D.2 in the setting of RFRBoost. For simplicity, we consider only the traditional ResNet structure where the random feature layer $f_t(x) = \tanh(B\Phi_t(x))$ only takes as input the previous layer of the ResNet and not the raw features. This corresponds to the weak feature transformation hypothesis class

$$\mathcal{G}_t = \{h \circ \Phi_{t-1} : h \in \mathcal{H}\}$$

where $\mathcal{H}$ is the set of simple residual blocks

$$\mathcal{H} = \{x \mapsto A\tanh(Bx) : \lambda_{\max}(A) \text{ and } \lambda_{\max}(B) \text{ bounded by } \lambda\}.$$

Here $\lambda_{\max}$ denotes the maximum singular value. We will use the properties that $\|A\|_2 \leq \lambda$, $\|A_k\|_1 \leq \lambda\sqrt{p}$, $\|A_k\|_\infty \leq \lambda$, $\|B_j\|_1 \leq \lambda\sqrt{D}$, and $\|B_j\|_\infty \leq \lambda$. Here $A_k$ denotes the $k$-th row of $A$, and $\|\cdot\|_p$ the $\ell_p$ vector norm or matrix spectral norm.

To prove Theorem 3.4, we need to compute the Rademacher complexities of RFRBoost, and verify that our class of weak learners satisfy all the assumptions of Theorem D.2. The only critical assumption to check is that $\sup_{g_t \in \mathcal{G}_t, x \in \mathcal{X}} \|g_t(x)\|_2$ is bounded. This is the case for our particular model class, since $\|g_t(x)\|_2 = \|A_t \tanh B_t \Phi_{t-1}(x)\|_2 \leq \|A_t\|_2 \|\tanh B_t \Phi_{t-1}(x)\|_2 \leq \lambda\sqrt{p}$, which follows from basic properties of vector and matrix norms.

The following lemma will prove useful for the computation of the Rademacher complexity of RFRBoost.

**Lemma D.3** (Allen-Zhu et al. (2019), Proposition A.12). *Let $\sigma : \mathbb{R} \to \mathbb{R}$ be a 1-Lipschitz function. Let $\mathcal{F}_1, ..., \mathcal{F}_m$ be sets of functions $\mathcal{X} \to \mathbb{R}$ and suppose for each $j \in [m]$ there exists a function $f_j \in \mathcal{F}_j$ satisfying $\sup_{x \in \mathcal{X}} |\sigma(f_j(x))| \leq R$. Then*

$$\mathcal{F} = \left\{x \mapsto \sum_{j=1}^m v_j \sigma(f_j(x)) : f_j \in \mathcal{F}_j, \|v\|_1 \leq C, v \in \mathbb{R}^m, \|v\|_\infty \leq A\right\}$$

*satisfies*

$$\mathfrak{R}(\mathcal{F}) \leq 2A\sum_{j=1}^m \mathfrak{R}(\mathcal{F}_j) + \mathcal{O}\left(\frac{CR\log m}{\sqrt{n}}\right).$$

### D.2. RFRBoost Rademacher Computations

The proof follows along similar lines as Suggala et al. (2020), however, our hypothesis class is larger and the proof has to be adjusted accordingly. Our proof additionally differs by bounding the hypothesis class by the largest singular values rather than $\ell_1$ norms, which we believe is more natural and leads to more representative bounds.

**Lemma D.4.** *Under the assumptions of Theorem 3.4, the Rademacher complexity $\mathfrak{R}(\mathcal{G}_t)$ satisfies*

$$\mathfrak{R}(\mathcal{G}_t) = \mathcal{O}\left(\frac{pD^{\frac{3}{2}}\log(D)t^{1-s}}{\sqrt{\widetilde{n}}}\right).$$

*Proof.* Using basic properties of the supremum, Hölder's inequality, Hoeffding's inequality, and Lemma D.3, we obtain the following:

$$\mathfrak{R}(\mathcal{G}_t) = \mathbb{E}_\rho\left[\sup_{g\in\mathcal{G}_t}\frac{1}{\widetilde{n}}\sum_{i=1}^{\widetilde{n}}\sum_{k=1}^{D}\rho_{i,k}g_k(x_{t,i})\right]$$

$$\leq \sum_{k=1}^{D}\mathbb{E}_\rho\left[\sup_{g\in\mathcal{G}_t}\frac{1}{\widetilde{n}}\sum_{i=1}^{\widetilde{n}}\rho_{i,k}g_k(x_{t,i})\right]$$

$$= \sum_{k=1}^{D}\mathbb{E}_\rho\left[\sup_{\substack{B\in\mathbb{R}^{p\times D}\\A\in\mathbb{R}^{D\times p}\\\lambda_{\max}(A),\lambda_{\max}(B)\leq\lambda_1}}\frac{1}{\widetilde{n}}\sum_{i=1}^{\widetilde{n}}\rho_{i,k}\langle A_k,\tanh(B\Phi_{t-1}(x_{t,i}))\rangle\right]$$

$$\leq 2\lambda_1 D\sum_{j=1}^{p}\mathbb{E}_\rho\left[\sup_{\substack{B\in\mathbb{R}^{p\times D}\\\lambda_{\max}(B)\leq\lambda_1}}\frac{1}{\widetilde{n}}\sum_{i=1}^{\widetilde{n}}\rho_{i,1}\langle B_j,\Phi_{t-1}(x_{t,i})\rangle\right] + D\mathcal{O}\left(\frac{\sqrt{p}\log p}{\sqrt{\widetilde{n}}}\right)$$

$$\leq 2\lambda_1^2 pD\sqrt{D}\mathbb{E}_\rho\left[\frac{1}{\widetilde{n}}\left\|\sum_{i=1}^{\widetilde{n}}\rho_{i,1}\Phi_{t-1}(x_{t,i})\right\|_\infty\right] + D\mathcal{O}\left(\frac{\sqrt{p}\log p}{\sqrt{\widetilde{n}}}\right)$$

$$\leq \frac{4\lambda_1^2 pD\sqrt{D}\log D}{\sqrt{\widetilde{n}}}\|\Phi_{t-1}(x_{t,i})\|_\infty + D\mathcal{O}\left(\frac{\sqrt{p}\log p}{\sqrt{\widetilde{n}}}\right)$$

$$\leq \frac{4\lambda_1^3 pD\sqrt{D}\log(D)ct^{1-s}}{(1-s)\sqrt{\widetilde{n}}} + D\mathcal{O}\left(\frac{\sqrt{p}\log(p)}{\sqrt{\widetilde{n}}}\right)$$

$$= \mathcal{O}\left(\frac{pD^{\frac{3}{2}}\log(D)t^{1-s}}{\sqrt{\widetilde{n}}}\right).$$

The second inequality follows from Lemma D.3 and the fact that $\tanh$ is bounded by $1$. The third inequality uses Hölder's inequality, and the fourth follows from Hoeffding's inequality for bounded random variables. Finally, the fifth inequality follows from the fact that $\|\Phi_{t-1}(x)\|_\infty \leq \sum_{r=1}^{t-1}\|g_r\|_\infty\eta_r \leq \frac{\lambda_1 ct^{1-s}}{1-s}$, derived from the recursive definition of $\Phi_t = \Phi_{t-1} + \eta_t g_t$. $\qquad\square$

**Lemma D.5.** *Under the assumptions of Theorem 3.4, the Rademacher complexity $\mathfrak{R}(\mathcal{W},\mathcal{G}_t)$ satisfies*

$$\mathfrak{R}(\mathcal{W},\mathcal{G}_t) = \mathcal{O}\left(\frac{pdD^{\frac{3}{2}}\log(D)t^{1-s}}{\sqrt{\widetilde{n}}}\right).$$

*Proof.* We proceed similarly as in the previous result, and obtain that

$$
\mathfrak{R}(\mathcal{W}, \mathcal{G}_t) = \mathbb{E}_\rho \left[ \sup_{\substack{W \in \mathcal{W} \\ g \in \mathcal{G}_t}} \frac{1}{\widetilde{n}} \sum_{i=1}^{\widetilde{n}} \sum_{j=1}^d \rho_{i,j} \left[ W^\top g(x_{t,i}) \right]_j \right]
$$

$$
\leq \sum_{j=1}^d \mathbb{E}_\rho \left[ \sup_{\substack{W \in \mathcal{W} \\ g \in \mathcal{G}_t}} \frac{1}{\widetilde{n}} \sum_{i=1}^{\widetilde{n}} \rho_{i,j} \left\langle W_j^\top, g(x_{t,i}) \right\rangle \right]
$$

$$
\leq^{(lemma)} 2\lambda_1 d \sum_{k=1}^D \mathbb{E}_\rho \left[ \sup_{g \in \mathcal{G}_t} \sum_{i=1}^{\widetilde{n}} \rho_{i,1} g_k(x_{t,i}) \right] + d\mathcal{O} \left( \frac{\lambda_1^2 \sqrt{pD} \log D}{\sqrt{\widetilde{n}}} \right)
$$

$$
\leq^{(prev\ result)} 2\lambda_1 d\mathcal{O} \left( \frac{pD^{\frac{3}{2}} \log(D) t^{1-s}}{\sqrt{\widetilde{n}}} \right) + d\mathcal{O} \left( \frac{\lambda_1^2 \sqrt{pD} \log D}{\sqrt{\widetilde{n}}} \right)
$$

$$
= \mathcal{O} \left( \frac{pdD^{\frac{3}{2}} \log(D) t^{1-s}}{\sqrt{\widetilde{n}}} \right).
$$

Here we additionally used the fact that $\sup_x |g_k(x)| = \sup_x |\langle A_k, \tanh(B\Phi_{t-1})(x)\rangle| \leq \lambda_1 \sqrt{p}$ for Lemma D.3. $\qquad\square$

We obtain Theorem 3.4 by combining Theorem D.2 with Lemma D.5 and Lemma D.4, and using the bound that $\sup_x \|g(x)\|_2 \leq \lambda \sqrt{p}$.

## E. Experimental Setup

This section provides additional details of the experimental setup, SWIM random features, evaluation procedures, baseline models, and hyperparameter ranges used in our numerical experiments.

### E.1. SWIM Random Features

In our experiments, we use SWIM random features (Bolager et al., 2023) for initializing the dense layer weights, as opposed to traditional i.i.d. initialization. Let $\mathcal{X}$ denote the input space and $\sigma$ an activation function. A SWIM layer is formally defined as follows:

**Definition E.1** (Bolager et al. (2023)). Let $\Phi(x) \in \mathbb{R}^D$ represent the features of a neural network for input $x \in \mathcal{X}$. Consider a dense layer $x \mapsto \sigma(Ax + b)$ to be added on top of $\Phi$, where $A \in \mathbb{R}^{H \times D}$, $b \in \mathbb{R}^H$, and $H$ is the number of neurons in the new layer. Let $(x_i^{(1)}, x_i^{(2)})_{i=1}^H \subset \mathcal{X} \times \mathcal{X}$ be pairs of training data points. Let

$$
A_i = c_2 \frac{\Phi(x_i^{(2)}) - \Phi(x_i^{(1)})}{\|\Phi(x_i^{(2)}) - \Phi(x_i^{(1)})\|^2}, \qquad b_i = -\langle A_i, \Phi(x_i^{(1)})\rangle - c_1, \tag{9}
$$

where $c_1$ and $c_2$ are fixed constants. We say that $x \mapsto \sigma(Ax + b)$ is a pair-sampled layer if the rows of the weight matrix $A$ and the biases $b$ are of the form (9).

The pairs of points $(x_i^{(1)}, x_i^{(2)})_{i=1}^H$ can be sampled uniformly or based on a training data-dependent sampling scheme. In our experiments, we adopt the gradient-based approach by Bolager et al. (2023). Pairs of data points are sampled at each layer $\Phi$ of the ResNet approximately proportional to

$$
q(x^{(1)}, x^{(2)}|\Phi) = \frac{\|f(x^{(1)}) - f(x^{(2)})\|}{\|\Phi(x^{(1)}) - \Phi(x^{(2)})\| + \epsilon}, \tag{10}
$$

where $\epsilon > 0$ and $f$ is the true target function, i.e., the mapping $x_i$ to the true label $y_i = f(x_i)$. Specifically, to avoid quadratic time complexity in dataset size $n$, we first consider the $n$ points $x_i^{(1)}$, $i = 1, ..., n$, and then uniformly generate $n$ offset points $x_i^{(2)} = x_{i+j_i}^{(1)}$. We then sample $H$ points from these proportional to $q(\cdot, \cdot|\Phi)$. This ensures the sampling procedure remains linear with respect to dataset size $n$.

The constants $c_1$ and $c_2$ are chosen such that the pre-activated neurons are symmetric about the input point $(\Phi(x_j^{(1)}) + \Phi(x_j^{(2)}))/2$. For general inputs, the pre-activation of neuron $j$ in a sampled layer is

$$(A\Phi(x) + b)_j = \langle A_j, \Phi(x) \rangle - \langle A_j, \Phi(x_j^{(1)}) \rangle - c_1$$
$$= c_2 \frac{\langle \Phi(x_j^{(2)}) - \Phi(x_j^{(1)}), \Phi(x) - \Phi(x_j^{(1)}) \rangle}{\|\Phi(x_j^{(2)}) - \Phi(x_j^{(1)})\|^2} - c_1,$$

and specifically, when applied to $x_j^{(2)}$ and $x_j^{(1)}$, we get

$$(A\Phi(x_j^{(1)}) + b)_j = -c_1,$$

$$(A\Phi(x_j^{(2)}) + b)_j = c_2 - c_1.$$

Thus, as suggested by Bolager et al. (2023), setting $c_2 = 2c_1$ centers the pre-activation symmetrically about the mean $(\Phi(x_i^{(1)}) + \Phi(x_i^{(2)}))/2$. Intuitively, this facilitates the creation of a decision boundary between the two chosen points. The constant $c_2$ is what we refer to as the *SWIM scale* in the hyperparameter section.

### E.2. Model Architectures and Hyperparameter Ranges for OpenML Tasks

We detail the hyperparameter ranges used when tuning all baseline models for the 91 regression and classification tasks of the curated OpenML benchmark suite. The hyperparameter tuning was conducted using the Bayesian optimization library Optuna (Akiba et al., 2019), with 100 trials per outer fold, model, and dataset, using an inner 5-fold cross validation. The specific ranges for each model are presented in Table 4.

*Table 4.* Hyperparameter ranges for OpenML experiments

| MODEL | HYPERPARAMETER | RANGE |
| --- | --- | --- |
| E2E MLP RESNET | N_LAYERS | 1 TO 10 |
| | LR | $10^{-6}$ TO $10^{-1}$ (LOG SCALE) |
| | END_LR_FACTOR | 0.01 TO 1.0 (LOG SCALE) |
| | N_EPOCHS | 10 TO 50 (LOG SCALE) |
| | WEIGHT_DECAY | $10^{-6}$ TO $10^{-3}$ (LOG SCALE) |
| | BATCH_SIZE | 128, 256, 384, 512 |
| | HIDDEN_DIM | 16 TO 512 (LOG SCALE) |
| | FEATURE_DIM | FIXED AT 512 |
| RFRBOOST | N_LAYERS | 1 TO 10 (LOG SCALE) |
| | L2_LINPRED | $10^{-5}$ TO 10 (LOG SCALE) |
| | L2_GHAT | $10^{-5}$ TO 10 (LOG SCALE) |
| | BOOST_LR | 0.1 TO 1.0 (LOG SCALE) |
| | SWIM_SCALE | 0.25 TO 2.0 |
| | FEATURE_DIM | FIXED AT 512 |
| RFNN | L2_REG | $10^{-5}$ TO 10 (LOG SCALE) |
| | FEATURE_DIM | 16 TO 512 (LOG SCALE) |
| | SWIM_SCALE | 0.25 TO 2.0 |
| RIDGE/LOGISTIC REGRESSION | L2_REG | $10^{-5}$ TO 10 (LOG SCALE) |
| XGBOOST | ALPHA | $10^{-5}$ TO $10^{-2}$ (LOG SCALE) |
| | LAMBDA | $10^{-3}$ TO 100 (LOG SCALE) |
| | LEARNING_RATE | 0.01 TO 0.5 (LOG SCALE) |
| | N_ESTIMATORS | 50 TO 1000 (LOG SCALE) |
| | MAX_DEPTH | 1 TO 10 |

For the E2E MLP ResNet, "hidden_dim" refers to the dimension $D$ of the ResNet features $\Phi(x) \in \mathbb{R}^D$, and the feature dimension denotes the number of neurons in the residual block (also known as bottleneck size, which was also fixed for

RFRBoost). The residual blocks are structured sequentially as follows: [dense(hidden_dim, feature_dim), batchnorm, relu, dense(feature_dim, hidden_dim), batchnorm]. An additional upscaling layer was used to match the input dimension to the "hidden_dim".

We used the Aeon library (Middlehurst et al., 2024) to generate the critical difference diagrams in Section 4.1. Below, we provide full dataset-wise results in Tables 5 to 7 for each model and dataset used in the evaluation, averaged across all 5 folds. Figures 5 and 6 visualize these results using scatter plots overlaid with box-and-whiskers plots, providing insight into the distribution and variability of performance across datasets. We refer the reader to the critical difference diagrams for a rigorous statistical comparison of model rankings across all datasets.

*Table 5.* Test classification scores on OpenML datasets.

| Dataset ID | Logistic Regression | RFRBoost | RFNN | E2E MLP ResNet | XGBoost |
|---|---|---|---|---|---|
| 3 | 0.974 (0.005) | 0.996 (0.002) | 0.989 (0.004) | 0.996 (0.001) | **0.997** (0.002) |
| 6 | 0.768 (0.005) | 0.912 (0.008) | 0.890 (0.009) | **0.944** (0.009) | 0.903 (0.015) |
| 11 | 0.878 (0.033) | **0.981** (0.006) | 0.979 (0.004) | 0.934 (0.042) | 0.931 (0.031) |
| 14 | 0.812 (0.011) | **0.844** (0.017) | 0.804 (0.018) | 0.828 (0.020) | 0.830 (0.008) |
| 15 | 0.964 (0.016) | **0.966** (0.011) | 0.963 (0.017) | 0.964 (0.015) | 0.960 (0.012) |
| 16 | 0.948 (0.004) | **0.974** (0.009) | 0.945 (0.008) | 0.973 (0.007) | 0.954 (0.016) |
| 18 | 0.728 (0.010) | **0.739** (0.010) | 0.727 (0.010) | **0.739** (0.007) | 0.721 (0.017) |
| 22 | 0.815 (0.012) | 0.837 (0.004) | 0.812 (0.011) | **0.840** (0.017) | 0.795 (0.012) |
| 23 | 0.512 (0.054) | 0.551 (0.038) | **0.553** (0.044) | 0.532 (0.035) | 0.547 (0.022) |
| 28 | 0.970 (0.005) | **0.991** (0.003) | 0.980 (0.006) | 0.989 (0.005) | 0.978 (0.004) |
| 29 | 0.854 (0.017) | 0.854 (0.024) | 0.845 (0.024) | 0.852 (0.027) | **0.862** (0.026) |
| 31 | 0.761 (0.023) | **0.762** (0.028) | 0.756 (0.028) | 0.743 (0.039) | 0.744 (0.031) |
| 32 | 0.946 (0.002) | **0.993** (0.003) | 0.990 (0.002) | 0.992 (0.001) | 0.986 (0.003) |
| 37 | 0.762 (0.020) | 0.758 (0.027) | 0.760 (0.032) | **0.767** (0.025) | 0.762 (0.034) |
| 38 | 0.968 (0.005) | 0.979 (0.005) | 0.977 (0.004) | 0.977 (0.004) | **0.988** (0.004) |
| 44 | 0.931 (0.007) | 0.950 (0.004) | 0.936 (0.009) | 0.941 (0.009) | **0.952** (0.002) |
| 50 | 0.983 (0.006) | 0.983 (0.010) | 0.980 (0.010) | 0.976 (0.016) | **0.995** (0.007) |
| 54 | 0.801 (0.026) | **0.849** (0.018) | 0.829 (0.021) | 0.843 (0.015) | 0.772 (0.019) |
| 151 | 0.760 (0.015) | 0.796 (0.014) | 0.784 (0.015) | 0.792 (0.010) | **0.847** (0.013) |
| 182 | 0.861 (0.009) | 0.909 (0.007) | 0.908 (0.007) | 0.911 (0.009) | **0.913** (0.006) |
| 188 | 0.641 (0.057) | **0.664** (0.049) | 0.643 (0.031) | 0.635 (0.038) | 0.652 (0.029) |
| 307 | 0.838 (0.012) | **0.993** (0.002) | 0.972 (0.016) | 0.989 (0.010) | 0.924 (0.031) |
| 458 | **0.994** (0.000) | 0.994 (0.007) | 0.993 (0.006) | 0.988 (0.013) | 0.990 (0.006) |
| 469 | 0.187 (0.023) | 0.188 (0.028) | **0.196** (0.021) | 0.193 (0.028) | 0.178 (0.031) |
| 1049 | 0.908 (0.012) | 0.910 (0.013) | 0.904 (0.008) | 0.907 (0.014) | **0.919** (0.009) |
| 1050 | 0.894 (0.013) | 0.889 (0.010) | **0.898** (0.010) | 0.892 (0.006) | 0.890 (0.007) |
| 1053 | 0.808 (0.009) | 0.807 (0.011) | 0.806 (0.012) | **0.810** (0.011) | 0.809 (0.014) |
| 1063 | 0.841 (0.023) | **0.845** (0.039) | 0.833 (0.024) | 0.841 (0.020) | 0.839 (0.021) |

*Table 6.* Test classification scores on OpenML datasets.

| Dataset ID | Logistic Regression | RFRBoost | RFNN | E2E MLP ResNet | XGBoost |
|---|---|---|---|---|---|
| 1067 | 0.853 (0.019) | 0.855 (0.018) | **0.858** (0.015) | 0.855 (0.016) | 0.853 (0.012) |
| 1068 | 0.925 (0.008) | 0.927 (0.016) | 0.928 (0.009) | 0.928 (0.010) | **0.935** (0.012) |
| 1461 | 0.902 (0.008) | 0.902 (0.005) | **0.904** (0.009) | 0.900 (0.006) | 0.903 (0.005) |
| 1462 | 0.988 (0.005) | **1.000** (0.000) | **1.000** (0.000) | **1.000** (0.000) | 0.997 (0.004) |
| 1464 | 0.766 (0.039) | 0.791 (0.037) | **0.794** (0.046) | 0.769 (0.048) | 0.783 (0.043) |
| 1475 | 0.478 (0.005) | 0.545 (0.010) | 0.534 (0.012) | 0.570 (0.005) | **0.601** (0.008) |
| 1480 | **0.715** (0.037) | 0.703 (0.017) | 0.703 (0.011) | 0.671 (0.033) | 0.684 (0.020) |
| 1486 | 0.943 (0.006) | 0.949 (0.003) | 0.946 (0.004) | 0.950 (0.006) | **0.959** (0.003) |
| 1487 | 0.935 (0.012) | 0.945 (0.012) | **0.945** (0.012) | 0.944 (0.009) | 0.940 (0.011) |
| 1489 | 0.748 (0.010) | 0.876 (0.015) | 0.864 (0.005) | **0.906** (0.005) | 0.903 (0.006) |
| 1494 | 0.873 (0.016) | **0.882** (0.016) | 0.868 (0.018) | 0.878 (0.015) | 0.873 (0.010) |
| 1497 | 0.701 (0.009) | 0.929 (0.005) | 0.907 (0.008) | 0.944 (0.007) | **0.998** (0.001) |
| 1510 | **0.977** (0.004) | 0.974 (0.008) | 0.974 (0.014) | 0.961 (0.023) | 0.961 (0.013) |
| 1590 | 0.850 (0.005) | 0.858 (0.010) | 0.856 (0.010) | 0.851 (0.009) | **0.865** (0.013) |
| 4534 | 0.938 (0.007) | 0.952 (0.010) | 0.949 (0.003) | **0.960** (0.009) | 0.956 (0.008) |
| 4538 | 0.478 (0.014) | 0.554 (0.017) | 0.513 (0.014) | 0.606 (0.014) | **0.660** (0.017) |
| 6332 | 0.735 (0.031) | 0.791 (0.024) | 0.763 (0.009) | 0.791 (0.023) | **0.796** (0.021) |
| 23381 | 0.626 (0.040) | **0.640** (0.030) | 0.624 (0.034) | 0.566 (0.048) | 0.612 (0.041) |
| 23517 | 0.505 (0.010) | **0.516** (0.009) | 0.502 (0.017) | 0.489 (0.013) | 0.500 (0.012) |
| 40499 | 0.997 (0.001) | 0.997 (0.001) | 0.992 (0.000) | **0.999** (0.001) | 0.985 (0.004) |
| 40668 | 0.752 (0.017) | 0.775 (0.014) | 0.759 (0.019) | 0.777 (0.010) | **0.800** (0.014) |
| 40701 | 0.867 (0.003) | 0.936 (0.002) | 0.923 (0.004) | 0.951 (0.009) | **0.953** (0.003) |
| 40966 | 0.994 (0.005) | **0.999** (0.002) | 0.989 (0.007) | 0.997 (0.002) | 0.984 (0.016) |
| 40975 | 0.932 (0.007) | 0.997 (0.004) | 0.977 (0.006) | **0.999** (0.001) | 0.994 (0.004) |
| 40982 | 0.718 (0.028) | 0.752 (0.016) | 0.758 (0.024) | 0.755 (0.028) | **0.800** (0.018) |
| 40983 | 0.971 (0.003) | 0.986 (0.002) | 0.987 (0.003) | **0.988** (0.004) | 0.983 (0.003) |
| 40984 | 0.901 (0.009) | **0.934** (0.012) | 0.930 (0.007) | 0.927 (0.011) | 0.929 (0.011) |
| 40994 | **0.961** (0.018) | 0.957 (0.019) | 0.957 (0.019) | 0.946 (0.015) | 0.944 (0.021) |
| 41027 | 0.685 (0.004) | 0.812 (0.007) | 0.798 (0.012) | **0.844** (0.007) | 0.838 (0.006) |

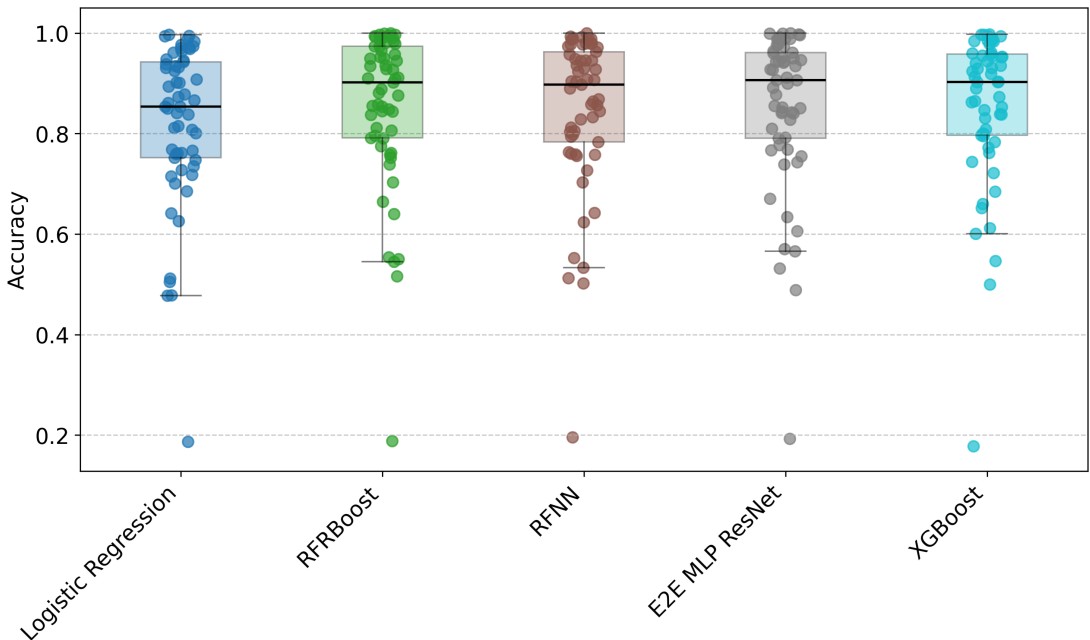

*Figure 5.* Scatter and box-and-whiskers plot of classification accuracy across all OpenML datasets. Each dot corresponds to a single dataset, and box plots summarize the distribution for each model.

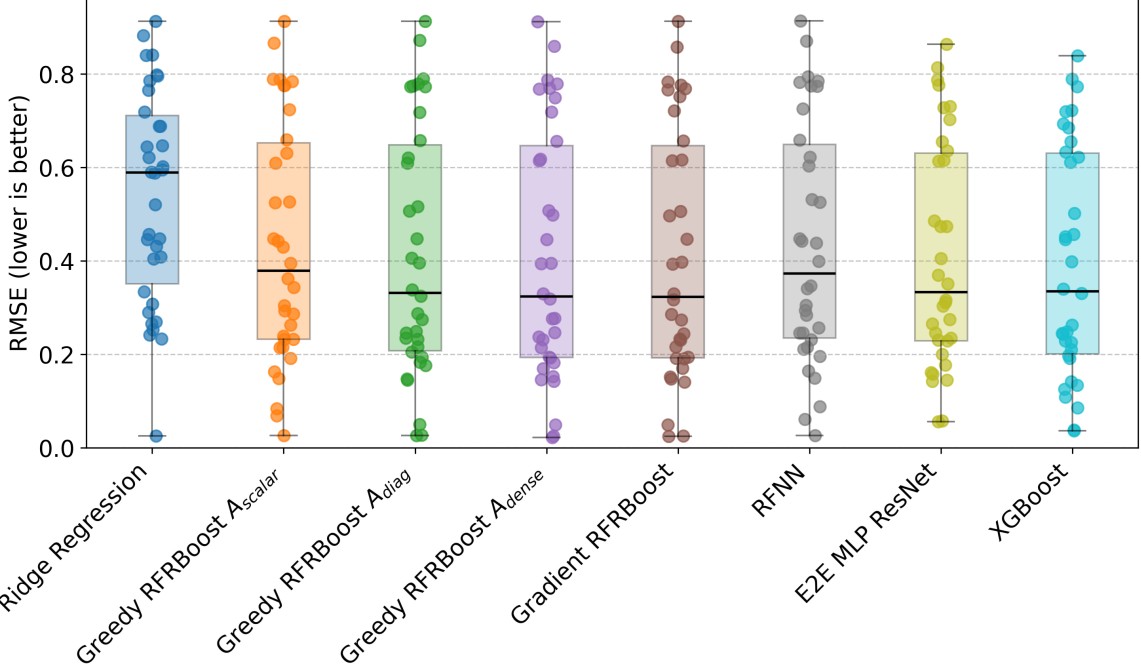

*Figure 6.* Scatter and box-and-whiskers plot of regression RMSE across all OpenML datasets. Each dot corresponds to a single dataset, and box plots summarize the distribution for each model.

*Table 7.* Test RMSE scores on OpenML regression datasets.

| Dataset ID | Ridge Regression | Greedy RFRBoost $A_{scalar}$ | Greedy RFRBoost $A_{diag}$ | Greedy RFRBoost $A_{dense}$ | Gradient RFRBoost | RFNN | E2E MLP ResNet | XGBoost |
|---|---|---|---|---|---|---|---|---|
| 41021 | 0.233 (0.012) | 0.233 (0.013) | 0.232 (0.013) | 0.231 (0.012) | **0.231** (0.011) | 0.232 (0.012) | 0.265 (0.011) | 0.249 (0.011) |
| 44956 | 0.647 (0.020) | 0.631 (0.018) | 0.621 (0.019) | 0.615 (0.019) | 0.616 (0.020) | 0.622 (0.022) | **0.614** (0.023) | 0.633 (0.020) |
| 44957 | 0.688 (0.033) | 0.362 (0.026) | 0.246 (0.017) | 0.238 (0.011) | 0.233 (0.013) | 0.347 (0.021) | 0.316 (0.026) | **0.197** (0.015) |
| 44958 | 0.595 (0.010) | 0.239 (0.019) | 0.205 (0.010) | 0.193 (0.016) | 0.191 (0.027) | 0.246 (0.022) | 0.246 (0.025) | **0.039** (0.011) |
| 44959 | 0.589 (0.026) | 0.304 (0.007) | 0.275 (0.013) | 0.277 (0.012) | 0.286 (0.018) | 0.305 (0.015) | 0.311 (0.011) | **0.245** (0.026) |
| 44960 | 0.290 (0.028) | 0.069 (0.013) | 0.050 (0.005) | 0.049 (0.005) | 0.049 (0.006) | 0.088 (0.024) | 0.200 (0.027) | **0.037** (0.007) |
| 44962 | **0.446** (0.063) | 0.448 (0.062) | 0.448 (0.062) | 0.446 (0.063) | 0.447 (0.062) | 0.447 (0.063) | 0.474 (0.069) | 0.452 (0.067) |
| 44963 | 0.841 (0.011) | 0.784 (0.005) | 0.773 (0.020) | 0.749 (0.008) | 0.752 (0.005) | 0.783 (0.013) | 0.703 (0.017) | **0.694** (0.010) |
| 44964 | 0.521 (0.017) | 0.443 (0.018) | 0.406 (0.013) | 0.395 (0.012) | 0.398 (0.009) | 0.443 (0.014) | 0.369 (0.005) | **0.340** (0.010) |
| 44965 | 0.883 (0.066) | 0.867 (0.071) | 0.873 (0.077) | 0.860 (0.075) | 0.858 (0.070) | 0.871 (0.077) | 0.864 (0.075) | **0.839** (0.081) |
| 44966 | 0.719 (0.082) | 0.724 (0.085) | **0.718** (0.084) | 0.719 (0.082) | 0.721 (0.080) | 0.726 (0.075) | 0.731 (0.079) | 0.720 (0.080) |
| 44967 | 0.786 (0.065) | 0.788 (0.071) | 0.790 (0.068) | 0.788 (0.065) | **0.784** (0.067) | 0.795 (0.069) | 0.814 (0.045) | 0.789 (0.056) |
| 44969 | 0.409 (0.011) | 0.084 (0.027) | 0.027 (0.007) | **0.022** (0.002) | 0.025 (0.002) | 0.061 (0.022) | 0.058 (0.015) | 0.086 (0.003) |
| 44970 | 0.645 (0.042) | 0.609 (0.044) | 0.610 (0.044) | 0.618 (0.035) | 0.615 (0.040) | **0.603** (0.045) | 0.616 (0.029) | 0.622 (0.043) |
| 44971 | 0.840 (0.028) | 0.789 (0.018) | 0.779 (0.022) | 0.780 (0.017) | 0.777 (0.021) | 0.785 (0.024) | 0.729 (0.009) | **0.685** (0.015) |
| 44972 | 0.796 (0.036) | 0.776 (0.028) | 0.775 (0.030) | 0.768 (0.026) | 0.767 (0.030) | 0.774 (0.026) | 0.777 (0.029) | **0.722** (0.012) |
| 44973 | 0.602 (0.007) | 0.287 (0.013) | 0.235 (0.008) | 0.194 (0.007) | 0.194 (0.004) | 0.284 (0.016) | **0.177** (0.007) | 0.244 (0.008) |
| 44974 | 0.457 (0.014) | 0.214 (0.013) | 0.184 (0.015) | 0.153 (0.008) | 0.152 (0.007) | 0.211 (0.012) | 0.158 (0.012) | **0.126** (0.012) |
| 44975 | 0.026 (0.008) | 0.027 (0.008) | 0.026 (0.008) | **0.026** (0.008) | 0.026 (0.008) | 0.026 (0.008) | 0.056 (0.012) | 0.109 (0.003) |
| 44976 | 0.270 (0.004) | 0.192 (0.007) | 0.177 (0.009) | 0.170 (0.005) | 0.170 (0.006) | 0.196 (0.009) | **0.161** (0.006) | 0.192 (0.007) |
| 44977 | 0.590 (0.021) | 0.525 (0.019) | 0.517 (0.016) | 0.508 (0.019) | 0.506 (0.021) | 0.525 (0.018) | 0.486 (0.020) | **0.446** (0.022) |
| 44978 | 0.242 (0.016) | 0.149 (0.005) | 0.145 (0.007) | 0.146 (0.006) | 0.148 (0.007) | 0.149 (0.006) | 0.142 (0.004) | **0.134** (0.011) |
| 44979 | 0.252 (0.015) | 0.163 (0.015) | 0.148 (0.012) | 0.142 (0.010) | **0.140** (0.010) | 0.165 (0.013) | 0.145 (0.011) | 0.142 (0.011) |
| 44980 | 0.766 (0.022) | 0.430 (0.008) | 0.324 (0.007) | 0.318 (0.008) | 0.317 (0.007) | 0.438 (0.022) | **0.275** (0.010) | 0.457 (0.031) |
| 44981 | 0.914 (0.017) | 0.913 (0.016) | 0.913 (0.017) | 0.912 (0.017) | 0.913 (0.017) | 0.914 (0.017) | 0.636 (0.015) | **0.611** (0.006) |
| 44983 | 0.431 (0.004) | 0.263 (0.012) | 0.250 (0.013) | 0.247 (0.016) | 0.244 (0.014) | 0.257 (0.010) | 0.235 (0.019) | **0.229** (0.014) |
| 44984 | 0.689 (0.023) | 0.660 (0.023) | 0.658 (0.024) | 0.656 (0.022) | 0.657 (0.023) | 0.659 (0.023) | 0.656 (0.023) | **0.656** (0.023) |
| 44987 | 0.334 (0.035) | 0.233 (0.020) | 0.194 (0.024) | **0.183** (0.026) | 0.192 (0.036) | 0.246 (0.019) | 0.229 (0.057) | 0.211 (0.020) |
| 44989 | 0.308 (0.010) | 0.293 (0.009) | 0.287 (0.006) | 0.276 (0.012) | 0.274 (0.013) | 0.295 (0.009) | 0.304 (0.016) | **0.263** (0.005) |
| 44990 | 0.404 (0.009) | 0.395 (0.009) | 0.396 (0.008) | 0.395 (0.009) | **0.393** (0.009) | 0.399 (0.010) | 0.405 (0.008) | 0.399 (0.008) |
| 44993 | 0.799 (0.017) | 0.776 (0.020) | 0.773 (0.017) | 0.770 (0.018) | **0.769** (0.019) | 0.775 (0.017) | 0.787 (0.018) | 0.773 (0.019) |
| 44994 | 0.265 (0.016) | 0.215 (0.011) | 0.216 (0.009) | **0.215** (0.007) | 0.216 (0.008) | 0.216 (0.009) | 0.231 (0.013) | 0.225 (0.010) |
| 45012 | 0.448 (0.027) | 0.343 (0.020) | 0.339 (0.019) | 0.330 (0.018) | **0.330** (0.017) | 0.341 (0.016) | 0.351 (0.022) | 0.331 (0.021) |
| 45402 | 0.621 (0.023) | 0.526 (0.024) | 0.507 (0.017) | 0.499 (0.017) | 0.497 (0.019) | 0.532 (0.019) | **0.474** (0.030) | 0.502 (0.029) |

## E.3. Point Cloud Separation Task

For the point cloud separation task, we use the same model structure as in the OpenML experiments, but fix the number of ResNet layers to 3 residual blocks. We additionally fix the activation to tanh and feature dimension to 512 for all models. Further architectural details and learning rate annealing strategies are consistent with those described in Section 4.1 and Appendix E.2. The SWIM scale was fixed at 1.0 for all random feature models. We use 5,000 randomly sampled data points for training and validation, and leave the remaining 5,000 for the final test set.

The hyperparameter configurations for RFRBoost, E2E MLP ResNet, RFNN, and logistic regression are detailed below in Table 8.

*Table 8.* Hyperparameter ranges for the point cloud separation task.

| MODEL | HYPERPARAMETER | VALUES |
|---|---|---|
| RFRBOOST | L2_CLS | [1, 1E-1, 1E-2, 1E-3, 1E-4, 1E-5] |
| | L2_GHAT | 1E-4 |
| | HIDDEN_DIM | 2 |
| | FEATURE_DIM | 512 |
| | BOOST_LR | 1.0 |
| | USE_BATCHNORM | TRUE |
| RFNN | L2_CLS | [1, 1E-1, 1E-2, 1E-3, 1E-4, 1E-5] |
| | FEATURE_DIM | 512 |
| LOGISTIC REGRESSION | L2 | [1, 1E-1, 1E-2, 1E-3, 1E-4] |
| E2E MLP RESNET | LR | [1E-1, 1E-2, 1E-3, 1E-4, 1E-5] |
| | HIDDEN_DIM | 2 |
| | FEATURE_DIM | 512 |
| | N_EPOCHS | 30 |
| | END_LR_FACTOR | 0.01 |
| | WEIGHT_DECAY | 1E-5 |
| | BATCH_SIZE | 128 |

## E.4. SWIM vs i.i.d. Ablation Study

In this section, we present a small ablation study comparing SWIM random features to standard i.i.d. Gaussian random features across OpenML regression and classification tasks. The evaluation procedure is the same as presented in Section 4.1, the only change being that the SWIM scale parameter is replaced by an i.i.d. scale parameter, ranging logarithmically from 0.1 to 10 for all models. Pairwise comparisons for each model variant are visualized in Figures 7 to 8, where each point represents a dataset and compares the performance of SWIM and i.i.d. features. Summary statistics, including average RMSE/accuracy, number of dataset-wise wins, and average training time, are reported in Tables 9 to 10.

Overall, performance differences between SWIM and i.i.d. features are small but consistent. For regression, SWIM-based models achieve lower mean RMSE across all variants, and win on 22 out of 34 datasets in the best case (Greedy RFRBoost $A_{\text{dense}}$). For classification, accuracy is nearly identical across methods, though i.i.d. features slightly outperform SWIM in the total number of wins (32 out of 56). SWIM-based methods typically incur slightly higher training time, reflecting the additional structure in their initialization. These results suggest that while SWIM can offer marginal gains in accuracy or RMSE, the benefits must be balanced against training efficiency.

*Table 9.* Comparison of SWIM vs i.i.d. feature initialization on OpenML regression tasks.

| MODEL | VARIANT | MEAN RMSE | WINS | FIT TIME (S) |
|---|---|---|---|---|
| RFNN | SWIM | 0.434 | 20 | 0.053 |
| | I.I.D. | 0.441 | 14 | 0.029 |
| GRADIENT RFRBOOST | SWIM | 0.408 | 18 | 1.688 |
| | I.I.D. | 0.415 | 16 | 1.816 |
| GREEDY RFRBOOST $A_{dense}$ | SWIM | 0.408 | 22 | 2.734 |
| | I.I.D. | 0.416 | 12 | 2.667 |
| GREEDY RFRBOOST $A_{diag}$ | SWIM | 0.415 | 19 | 1.631 |
| | I.I.D. | 0.415 | 15 | 1.490 |
| GREEDY RFRBOOST $A_{scalar}$ | SWIM | 0.434 | 22 | 1.024 |
| | I.I.D. | 0.443 | 12 | 0.720 |

*Table 10.* Comparison of SWIM vs i.i.d. feature initialization on OpenML classification tasks.

| MODEL | VARIANT | MEAN ACCURACY | WINS | FIT TIME (S) |
|---|---|---|---|---|
| RFNN | SWIM | 0.845 | 25 | 1.189 |
| | I.I.D. | 0.844 | 31 | 0.634 |
| GRADIENT RFRBOOST | SWIM | 0.853 | 32 | 2.519 |
| | I.I.D. | 0.850 | 24 | 2.350 |

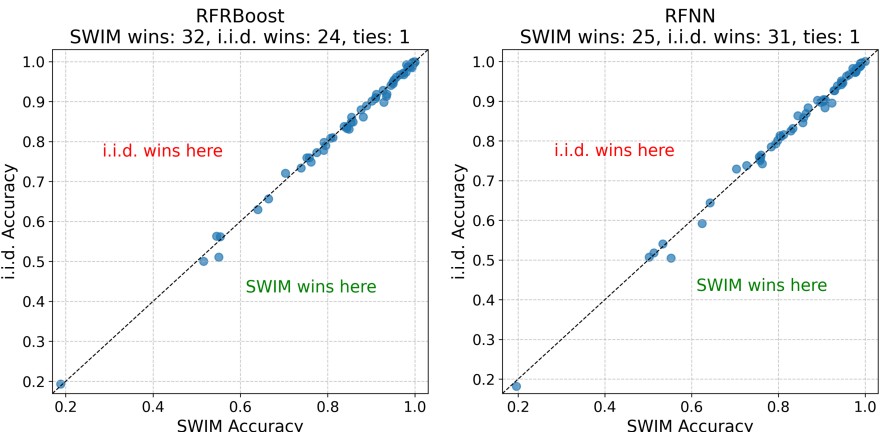

*Figure 7.* Comparison of classification accuracy using SWIM versus i.i.d. Gaussian random features across OpenML datasets. Each scatter plot compares accuracy on a per-dataset basis for different model variants. Points below the diagonal indicate better performance with SWIM.

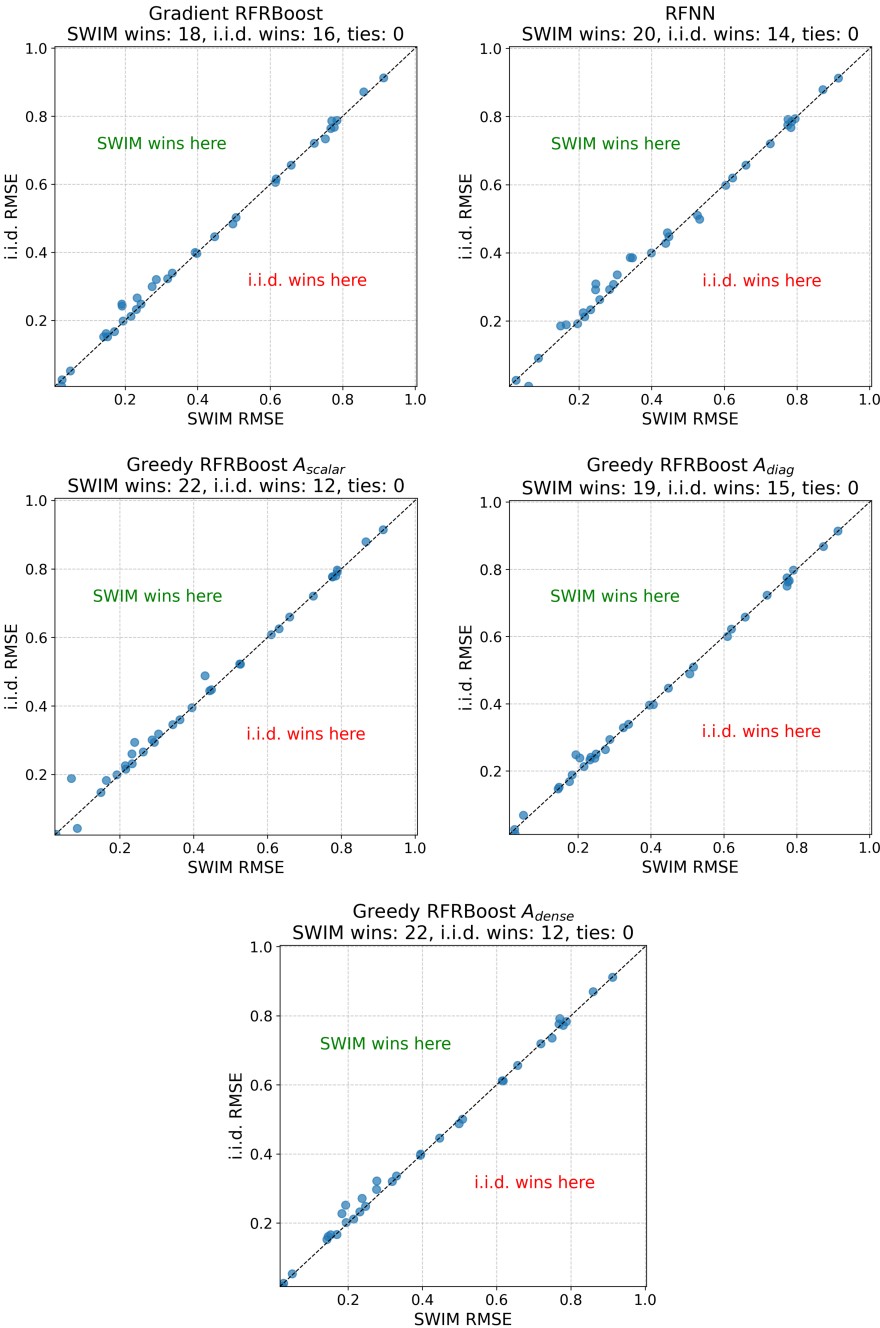

*Figure 8.* Comparison of RMSE using SWIM versus i.i.d. Gaussian random features across OpenML regression tasks. Each scatter plot compares RMSE per dataset for different model variants. Points above the diagonal indicate better performance with SWIM.

### E.5. Supplementary Larger-Scale Experiments

To further assess the scalability and performance of RFRBoost relative to baseline models, we conducted experiments on four larger datasets: two of the largest from the OpenML Benchmark suite used in this work (each with approximately 100k samples — one classification: ID 23517; one regression: ID 44975), and two widely-used benchmark datasets with approximately 500k samples (one classification: CoverType (Blackard, 1998); one regression: YearPredictionMSD (YPMSD) (Bertin-Mahieux, 2011)).

For these larger-scale evaluations, we employed a hold-out test set strategy. Hyperparameters for each model were selected via a grid search performed on a 20% validation split of the training data. The models considered are consistent with those in Section 4.1: E2E MLP ResNet, Gradient RFRBoost, Logistic/Ridge Regression, RFNN, and XGBoost. To analyze performance trends with increasing data availability, models were trained on exponentially increasing subsets of the training data, starting from 2048 samples ($2^{11}$) up to the maximum available training size for each dataset.

For the OpenML datasets, the test sets were defined as 33% (ID 23517, classification) and 11% (ID 44975, regression) of the total instances, resulting in a maximum training set size of approximately 64,000 instances ($2^{16}$). For YPMSD (regression), we adhered to the designated split: the first 463,715 examples for training and the subsequent 51,630 examples for testing. For CoverType (classification), we used the standard train-test split provided by the popular Python machine learning library Scikit-learn (Pedregosa et al., 2011), yielding 464,809 training and 116,203 test instances.

The hyperparameter grids used for each model in these experiments are detailed in Table 11. To account for variability, each experiment was repeated 5 times with different random seeds for each model. All experiments were carried out on a single NVIDIA RTX 6000 (Turing architecture) GPU.

*Table 11.* Hyperparameter grid for the larger-scale experiments.

| MODEL | HYPERPARAMETER | VALUES |
|---|---|---|
| RFRBOOST | L2_CLS | [1E-1, 1E-2, 1E-3, 1E-4, 1E-5] |
| | N_LAYERS | [1, 3, 6] |
| | FEATURE_DIM | 512 |
| | BOOST_LR | 1.0 |
| RFNN | L2 | [1E-1, 1E-2, 1E-3, 1E-4, 1E-5] |
| | FEATURE_DIM | 512 |
| LOGISTIC/RIDGE REGRESSION | L2 | [1E-1, 1E-2, 1E-3, 1E-4] |
| XGBOOST | LR | [0.1, 0.033, 0.01] |
| | N_ESTIMATORS | [250, 500, 1000] |
| | MAX_DEPTH | [1, 3, 6, 10] |
| | LAMBDA | 1.0 |
| E2E MLP RESNET | LR | [1E-1, 1E-2, 1E-3] |
| | N_EPOCHS | [10, 20, 30] |
| | FEATURE_DIM | 512 |
| | END_LR_FACTOR | 0.01 |
| | WEIGHT_DECAY | 1E-5 |
| | BATCH_SIZE | 256 |

The model training times and predictive performance (RMSE for regression, accuracy for classification) as a function of training set size are presented in Figure 9 and Figure 10, respectively. Regarding training times shown in Figure 9, Ridge/Logistic Regression and RFNNs consistently exhibit the fastest training, as expected. RFRBoost consistently trains faster than both XGBoost and the E2E MLP ResNets across all datasets. Notably, on the OpenML ID 23517 dataset, RFRBoost trains faster than the single-layer RFNN. This may be attributed to RFNNs fitting a potentially very wide random feature layer directly to the output targets (logits or regression values), which can be computationally intensive. In contrast, RFRBoost constructs a comparatively shallow representation layer-wise with random feature residual blocks, where only the final, potentially lower-dimensional, representation is mapped to the output targets. In some scenarios, this may lead to lower computational times.

Considering the predictive performance shown in Figure 10, RFRBoost consistently and significantly outperforms the

baseline single-layer RFNN across all datasets and training sizes, demonstrating its effectiveness in using depth for random feature models to increase performance. This improvement is particularly pronounced on the larger CoverType and YPMSD datasets, underscoring the benefit of RFRBoost's deep, layer-wise construction. However, in the larger data regimes E2E trained MLP ResNets and XGBoost generally achieve superior predictive performance. The OpenML ID 23517 dataset stands out, however, as RFRBoost and Logistic Regression not only exhibit lower variance but also achieve the best overall performance. Furthermore, we find that XGBoost significantly underperforms on OpenML ID 44975 compared to all other baseline models. Given RFRBoost's strong performance relative to its computational cost and its ability to construct meaningful deep representations, exploring its use as an initialization strategy for subsequent fine-tuning of end-to-end trained networks presents a promising avenue for future research, potentially further enhancing the performance of E2E models.

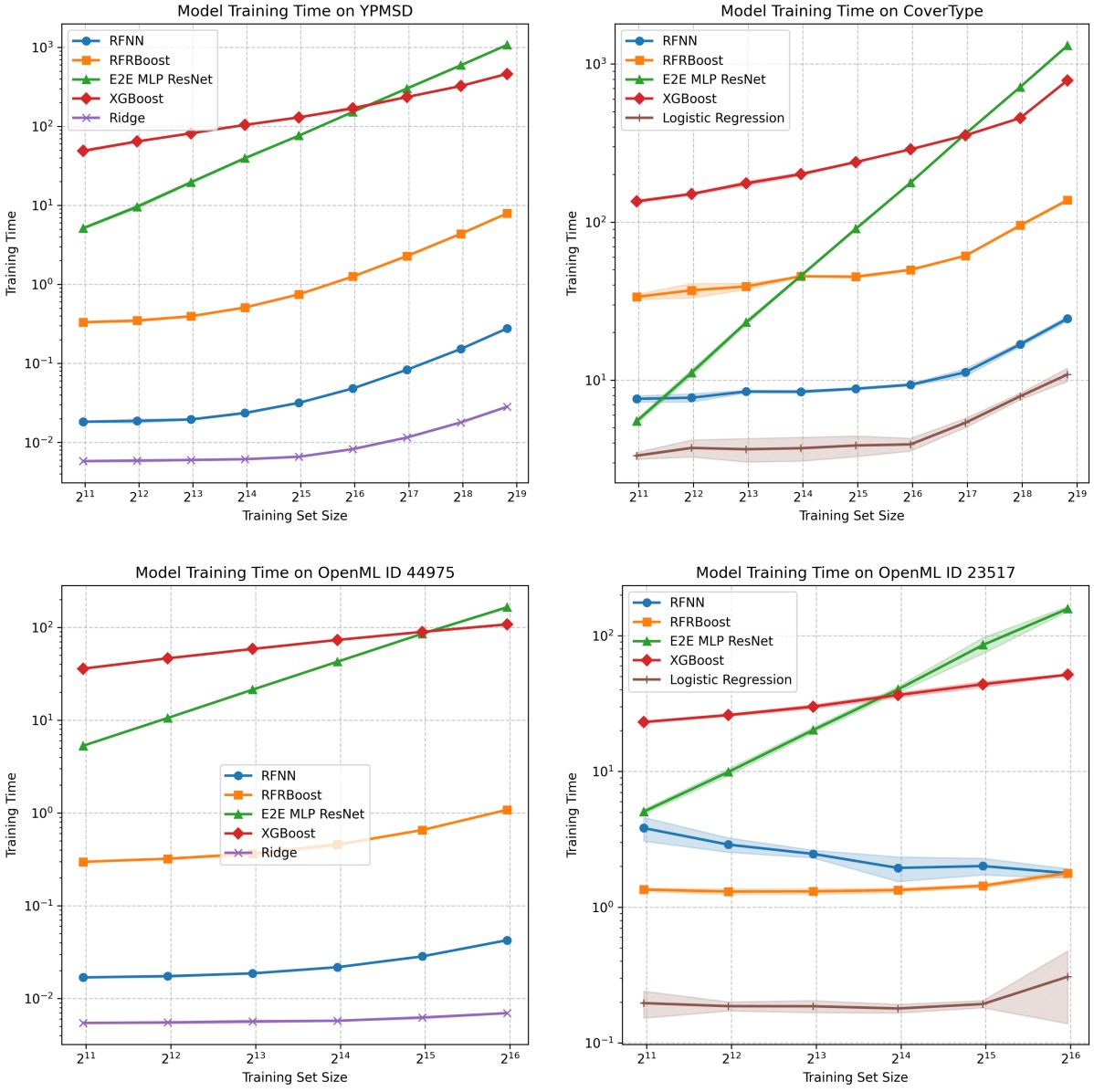

*Figure 9.* Model training time (seconds, log scale) versus training set size on larger datasets. Lines represent the mean training time over 5 runs, and shaded areas indicate 95% confidence intervals.

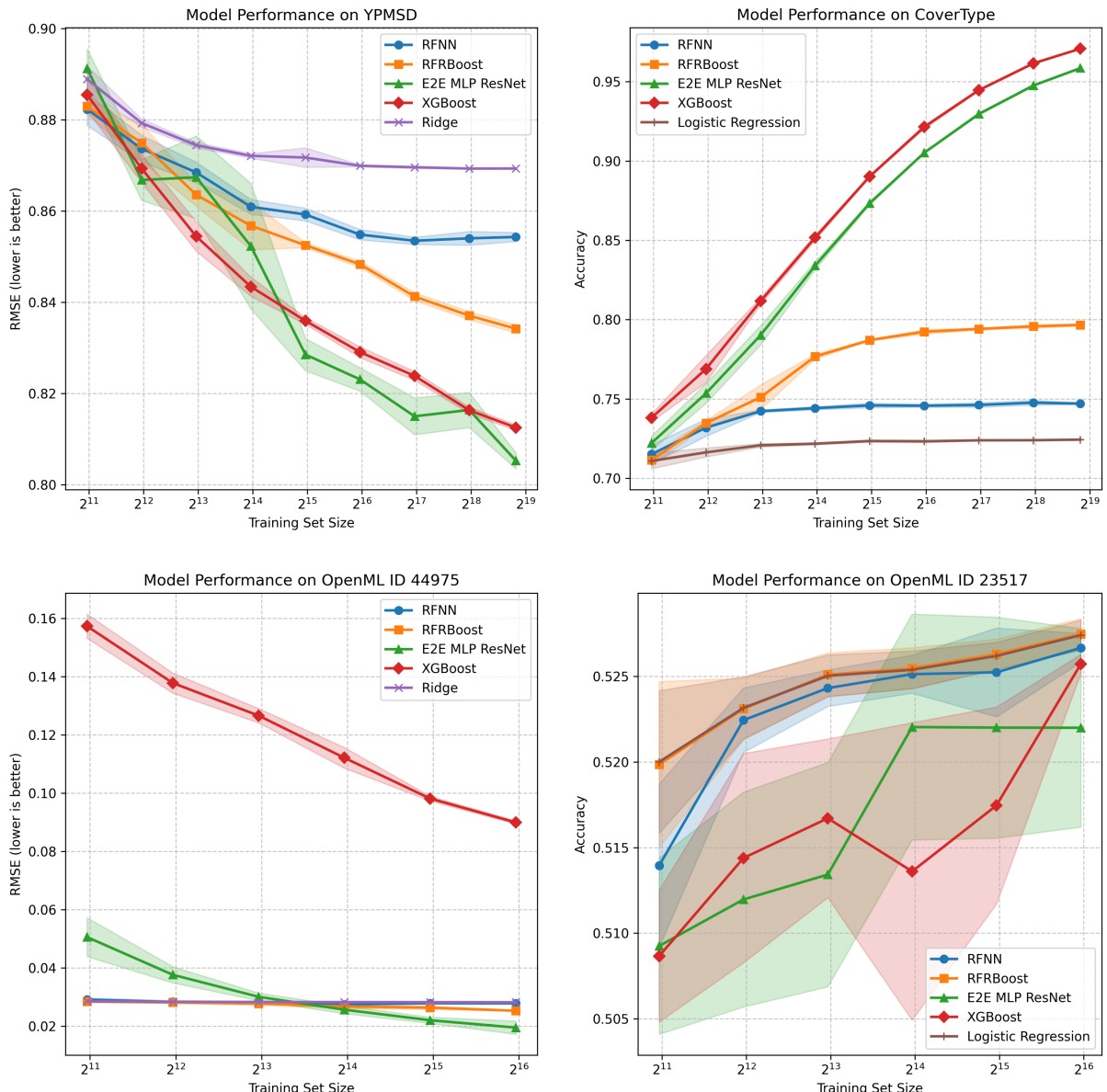

*Figure 10.* Model predictive performance (RMSE for regression, accuracy for classification) versus training set size on larger datasets. Lines represent the mean performance over 5 runs, and shaded areas indicate 95% confidence intervals.

