# OpenReview forum: "Random Feature Representation Boosting"
_ICML.cc/2025/Conference — ICML 2025 poster_

### Official Review · Reviewer_JjfD · 2025-03-11

**Overall Recommendation:** 3

**Summary:**

This paper introduces a Random Feature Representation Boosting method which uses random features at each layer and iteratively optimizes the functional gradient of the network representation. Experiments shows it outperforms traditional RFNNs and end-to-end trained MLP ResNets.

**Claims And Evidence:**

My question is what is the main differences between Random Feature Representation Boosting and Existing Functional Boosting, My understanding is the dense version of  Random Feature Representation Boosting is actually equal to Functional Gradient Boosting. And in latter evaluation, dense version performs better than the scalar and diag version.

**Essential References Not Discussed:**

N/A

**Experimental Designs Or Analyses:**

Can you provide insights into why XGBoost achieves a lower RMSE in regression tasks?

**Methods And Evaluation Criteria:**

Yes

**Other Comments Or Suggestions:**

N/A

**Other Strengths And Weaknesses:**

N/A

**Questions For Authors:**

N/A

**Relation To Broader Scientific Literature:**

This paper tries to extends boosting theory beyond function approximation to feature representation learning, integrating random feature methods with layer-wise boosting.

**Theoretical Claims:**

Theoretical results show that RFRBoost aligns with the theoretical framework of Generalized Boosting, satisfying its weak learning conditions and risk bound guarantees.

---

> ### Author Rebuttal · Authors · 2025-03-31
>
> Thank you for taking the time to review our work and for your comments. Below, we address your questions and clarify the novelty and key differences of our work.
>
> **Q: What are the main differences between Random Feature Representation Boosting and existing Functional Boosting? Are dense Random Feature Representation Boosting and Functional Gradient Boosting the same?**
>
> We appreciate the opportunity to clarify the distinctions. There are crucial differences between RFRBoost and what might be considered 'Existing Functional Gradient Boosting' (FGB), particularly the classical form used in methods like XGBoost.
>
>
> * **Boosting Target (Label Space vs. Feature Space):** Classical FGB, as outlined in our Section 2.1, typically operates in *label space*. It iteratively adds weak learners (like decision trees) to directly improve the model's prediction $F_t(x)$ by fitting the gradient of the loss with respect to the current prediction. In contrast, RFRBoost operates in the framework of *Gradient Representation Boosting* (GRB), detailed in Section 2.2. GRB boosts in *feature space*, iteratively building a deep residual-neural-network-like feature representation $\Phi_t(x)$ by adding residual blocks $g_t$ that approximate the functional gradient of the loss with respect to the neural network representation itself. The final prediction is then made by a **single** linear model $W^T \Phi_T(x)$ trained on the final representation. RFRBoost in the $\Delta_{dense}$ regime is not equivalent to classical gradient boosting, because only a single linear model is trained on the boosted ResNet feature representation, rather than an ensemble of such models as in traditional gradient boosting.
>
> * **Random Feature Residual Blocks $g_t$:** While the theory of GRB allows for general function classes $g_t$, RFRBoost specifically uses fixed random features, allowing for optimized computations and specialized theoretical results. More specifically, we have $g_t = A_t f_t$, where $f_t = f_t(x, \Phi_{t-1}(x))$ is generated using random weights, and only the linear map $A_t$ is trained using linear methods. This is fundamentally different from previous GRB methods that train bigger neural networks end-to-end as residual blocks using SGD. Our use of fixed random features is what enables efficient computational routines and allows for analytical solutions as per Theorems 3.1 and 3.2, while retaining the theoretical guarantees stemming from GRB (Theorem 3.3), and significantly improving performance (Section 4.1). A specific contribution within our gradient-greedy approach is the incorporation of a unit-norm constraint when learning $g_t$, differing from related literature. This ensures we correctly identify the optimal direction for the residual block in function space, which is crucial for the validity of the functional Taylor approximation of the risk. The special setting of random features is what allowed us to prove that this can be obtained by solving a constrained least squares problem (Theorem 3.2).
>
> **Q: Can you provide insights into why XGBoost achieves a lower RMSE in regression tasks?**
>
> It is challenging to pinpoint a single definitive reason why XGBoost achieves slightly lower RMSE than RFRBoost in our experiments, as RFRBoost and XGBoost operate under fundamentally different paradigms as explained above. (Note however that while XGBoost achieves a slightly lower RMSE than RFRBoost, it performs worse in terms of average rank.)
>
> XGBoost uses classical gradient boosting (Section 2.1) to construct a large ensemble, comprising hundreds or thousands of decision trees. It learns by fitting trees to the residual errors (gradients) in label/logit space. Its strengths lie in its ability to capture complex interactions via hierarchical, axis-aligned splits and regularization techniques tailored for trees.
>
> RFRBoost, conversely, uses gradient representation boosting (Section 2.2) to build a deep feature representation $\Phi_T(x)$ using random projections, followed by a single linear predictor. Its inductive bias stems from the nature of the random features used (e.g., dense projections with tanh activation in our experiments) and the layer-wise boosting process in feature space.
>
> Potential factors contributing to the observed difference on these specific benchmarks might be due to different inductive biases of the different approaches, ensembling vs using a single predictor, or difference in regularization of discrete trees versus continuous (logistic) regression targets.
>
> ---
>
> We hope this clarifies the methodological distinctions and provides context for the performance comparison. We aim to make this clearer in the revised paper.

---

> > ### Comment · Reviewer_JjfD · 2025-04-03
> >
> > Thanks for the clarification, I will raise my score to 3.

---

### Official Review · Reviewer_XEnQ · 2025-03-12

**Overall Recommendation:** 4

**Summary:**

This paper studies an area of "Extreeme learning" where the weights of a non-linear transformation are set without any gradient calculations. In this instances, a Residual Network connection is used to derive the math for setting the procedure in a deep randomized network for MSE prediction problems specifically. The results improve upon  prior approaches in the space and show tentatively close RMSE to XGboost.


--- Post Rebuttal ---

I still support this paper. I hope the larger scale results are done for camera ready, the preliminary results are encouraging.

**Claims And Evidence:**

The theory appears sound but I have not checked it in detail. The experimental methodology is very thorough in a way that we can ensure the comparisons being made are valid (Finally, someone uses the Wilcoxon signed rank test for model comparisons in an ML paper!). However, sub-sampling all datasets to <=  5k observations significantly weakens the strength of the conclusions.

**Essential References Not Discussed:**

I'm broadly familiar with basic extreme learning literature. I think discussing the related work in randomized/non-learned convolutional filters for CNNs would significantly improve the article and provide less enlightened readers with a better scope of the possible future works. In particular, Scattering networks perform the same task for CNNs specifically, and while I see no trivial path to including them in experimentation, I think they help establish the broader interest in this field of work:

See:
* Andreux, M., Angles, T., Exarchakis, G., Leonarduzzi, R., Rochette, G., Thiry, L., Zarka, J., Mallat, S., Andén, J., Belilovsky, E., Bruna, J., Lostanlen, V., Chaudhary, M., Hirn, M. J., Oyallon, E., Zhang, S., Cella, C., & Eickenberg, M. (2020). Kymatio: Scattering transforms in python. Journal of Machine Learning Research, 21(2012), 2012–2017.
* Bruna, J., & Mallat, S. (2013). Invariant Scattering Convolution Networks. IEEE Transactions on Pattern Analysis and Machine Intelligence, 35(8), 1872–1886. https://doi.org/10.1109/TPAMI.2012.230
* Cotter, F., & Kingsbury, N. G. (2017). Visualizing and improving scattering networks. In N. Ueda, S. Watanabe, T. Matsui, J.-T. Chien, & J. Larsen (Eds.), 27th IEEE International Workshop on Machine Learning for Signal Processing, MLSP 2017, Tokyo, Japan, September 25-28, 2017 (pp. 1–6). IEEE. https://doi.org/10.1109/MLSP.2017.8168136
* Mallat, S. (2012). Group Invariant Scattering. Communications on Pure and Applied Mathematics, 65(10), 1331–1398. https://doi.org/10.1002/cpa.21413
* Oyallon, E., Belilovsky, E., & Zagoruyko, S. (2017). Scaling the Scattering Transform: Deep Hybrid Networks. 2017 IEEE International Conference on Computer Vision (ICCV), 5619–5628. https://doi.org/10.1109/ICCV.2017.599
* Oyallon, E., Zagoruyko, S., Huang, G., Komodakis, N., Lacoste-Julien, S., Blaschko, M., & Belilovsky, E. (2019). Scattering Networks for Hybrid Representation Learning. IEEE Transactions on Pattern Analysis and Machine Intelligence, 41(9), 2208–2221. https://doi.org/10.1109/TPAMI.2018.2855738
* Trockman, A., Willmott, D., & Kolter, J. Z. (2022). Understanding the Covariance Structure of Convolutional Filters (No. arXiv:2210.03651). arXiv. https://doi.org/10.48550/arXiv.2210.03651

**Experimental Designs Or Analyses:**

I have checked all the experiments. The average reader ay not appreciate the difficulty of problem 4.2, and including results of resents of varying size trained with normal Adam to demonstrate how much computational/representational power to perform such separation would be beneficial in educating the less-informed.

While I understand the use of sub-sampling due to 91 datasets being used, including at least _some_ datasets with multiple larger sizes of varying orders of magnitude will majorly strengthen this work (e.g., benefit as datset sizes increase? Accuracy? Runtime? Lots of questions to answer).

**Methods And Evaluation Criteria:**

The scope of considered datasets is reasonable and the large number of datasets is good.

**Other Comments Or Suggestions:**

Some of the early exposition is a bit hard to follow and lacks explanation. E.g., what is a "law of the data"? The notation makes it unclear when something is a _function_ vs. a weight matrix (e.g., $f_t$ looks like a function but isn't!). Making the notation more consistent in something like "greek for functions" or some other style would help.

**Other Strengths And Weaknesses:**

See the above discussion.

**Questions For Authors:**

* Can you run XGBoost and your best proposed methods on more of the fulll-sized datasets and report per-dataset and aggregate results? In both accuracy and runtime would be very valuable.
* It seems like all experiments are MSE only, even though cross-entropy is stated to exist. Please clarify? Such problems should be presented as distinct groups rather than merged if both were done.
* If you saved the results, can you report the sensitivity of XGBoost and your method with respect to the hyper-parameters? e.g., are there "reliable defaults" for your method you observe empirically, or evidence of less fluctuation due to using the approach?

**Relation To Broader Scientific Literature:**

Not only are extrem learning machines useful in their own right, but they expose interesting scientific questions about network initialization strategies and the scope of what SGD at large can and is needed to learn and achieve good performance. I'm quite surprised that such high performance can be obtained without any active gradient steps or iterative calculations on a per-layer basis, which is fundamentally interesting in its own right.

**Theoretical Claims:**

I have not the time to check the theory, but the appendix seems to have step-by-step derivations. I'm particularly thankful for the inclusion of "Python code for this is" blocks in much of the appendix, a refreshing balance of math and code exposition. I'd encourage the authors to add this to the C.3. Categorial Cross-Entropy section as well for completeness and so that others may experiment with classification problems (indeed, it is a little saddening they were not included at the onset).

---

> ### Author Rebuttal · Authors · 2025-03-31
>
> We appreciate your valuable comments and thoughtful points raised, especially the positive feedback regarding experimental rigour. We address your points below.
>
> **Are results MSE only?:** We apologize if this was unclear. Regression tasks are reported in Table 1 and Figure 2, while classification tasks (cross-entropy loss) are reported in Table 2 and Figure 3. We will revise the text in Section 4.1 to explicitly state that the regression and classification results are presented separately. Additionally, we will also add scatter/box-and-whiskers plots to better display the variability of the models, an ablation study for SWIM vs iid features, and full dataset-wise results in the Appendix.
>
> RFRBoost supports any convex loss function, and the main body of the paper was developed with this in mind. The exact-greedy analytical solution are a special case for MSE loss, but the gradient-greedy algorithm of Sec 3.3 and Algo. 2 supports general convex loss function. More importantly, in the gradient-greedy case we argued why previous work on SGD-based representation boosting overlooked the crucial detail of constraining the functional norm of the residual block to unit norm. This is essential to obtain a valid functional Taylor approximation of the risk, and we proved in Theorem 3.2 how to efficiently calculate and incorporate this constraint when using random features. This step provably involves solving a constrained least squares problem (where the functional gradient vector depends on the convex loss), which is where the MSE confusion might have stemmed from.
>
>
> **References on CNNs and scattering networks:** We thank the referee for pointing out this related field of study. We will incorporate these topics into the related works section in the revised manuscript. We agree that there is no immediate way to include these in experimentation as they concern image data. However, this is an interesting and important direction for future research to build upon the foundational work presented in our paper.
>
>
> **Code Blocks:** We are glad you found the embedded Python code snippets helpful for clarity and reproducibility. Following your suggestion, we have added corresponding code examples for the gradient calculation in Appendix C.
>
> **Dataset Size and Scalability:**
> The decision to truncate datasets at 5000 observations stemmed primarily from computational resource constraints, balanced against the desire to experiment on a large number (91) of diverse datasets. Note that this threshold is substantially larger than the median dataset size (approx. 2400) within the OpenML Benchmark Suite. The primary computational bottlenecks were the E2E ResNet, and the rigorous nested 5-fold cross-validation procedure with Optuna (requiring 5*100 fits per dataset fold per model). Conducting such extensive evaluation on 91 datasets at their full size was computationally infeasible for us. However, we argue that evaluating performance on datasets of this scale is valuable, as this regime is highly relevant for many practical applications and is a common domain for RFNNs and extreme learning machines. Restricting dataset size is not uncommon in published papers (e.g. Bolager et al. 2023); we aimed for transparency by stating it directly in the main paper, rather than leaving this to the appendix.
>
> To further address scalability we will:
>
> * Add an analysis of the computational complexity of Algorithms 1 and 2, noting its scaling behaviour $O(NTD^2p^2d)$ for training.
>
> * Add experiments in the Appendix where RFRBoost and baselines are trained on larger datasets using increasing fractions of the data (e.g., 10k, 25k, 50k, 100k, 200k, 400k samples, Covertype, YP-MSD). These will be displayed in a plot with number of samples on the X axis, and results/time on Y, to demonstrate scaling behaviour.
>
> Scaling RFRBoost efficiently to datasets with millions of rows would likely require distributed computing paradigms such as MapReduce and implementation optimizations beyond the scope of this initial work, which we identify as an important direction for future research.
>
>
> **Terminology:** We have replaced the phrase "law of the data" with "distribution of the data". The original phrasing, while common in some areas of theoretical statistics, might be less familiar to the broader machine learning audience, and we have revised the manuscript accordingly.
>
>
>
>
> **Sensitivity of Parameters:**
> We observed that optimal hyperparameters were highly dataset-dependent for *all* models, including XGBoost, ridge/logistic regression, E2E MLP ResNets, and RFRBoost. This variability motivated our use of a rigorous nested 5-fold CV scheme coupled with Optuna for hyperparameter tuning, ensuring a fair comparison on all datasets. Anecdotally, we observed a general trend where RFRBoost performance tended to improve with increasing depth, whereas the optimal E2E MLP ResNets depth was more varied.
>
>
>
> ---
>
>
> We hope these responses adequately address your questions.

---

> > ### Comment · Reviewer_XEnQ · 2025-04-02
> >
> > Could you post in reply some of the large-scale results on one or two datasets?
> >
> > Are so many hyper-parameter steps _needed_ for the approach? Do you have the optuna plot/graph for performance over trials?

---

> > > ### Author Response · Authors · 2025-04-08
> > >
> > > **Larger datasets:** We have not yet been able to complete the additional larger-scale experiments. This took longer than anticipated, as the evaluation code needed rewriting for these additional experiments and thorough evaluation requires extensive, costly hyperparameter optimization.
> > >
> > > However, we can provide some preliminary full-scale results on the largest dataset in the used OpenML repository: id 23517 (classification, approx 100k samples).
> > >
> > > Model accuracy vs training set size: https://postimg.cc/HjWgK7Sj
> > >
> > > Model fit time vs training set size: https://postimg.cc/Pp3F0Nx8
> > >
> > > As seen in the plots, RFRBoost consistently outperforms RFNNs, E2E MLPs and XGBoost across all data sizes up to full size (we use the same fixed hold out test set). There is a healthy margin between traditional RFNNs and our deep random feature-boosted model at this scale, too. RFRBoost also exhibits lower variance across multiple runs compared to the other baseline models, especially E2E MLP ResNets. It is possible that E2E trained weights may eventually outperform random weights at some dataset size threshold. However, this is not yet the case here. We also emphasize that 55/91 of the datasets in our main evaluation were *not* truncated (being below threshold), and truncation does not impact the rigour of the evaluation procedure and statistical tests. RFNNs are often used in medium-sized data regimes, making our rigorous experiments highly relevant. We will nevertheless include some additional results on even larger-scale datasets in our revised manuscript, but we reiterate that this is not the main contribution of the paper. Rather, the main contribution is our novel methodology for constructing deep residual RFNNs for arbitrary random features and rigorous validation.
> > >
> > > This training time plot interestingly shows that RFRBoost can train faster than a single-layer RFNN. This is because RFNNs fit the wide random feature layer directly to logits, which is computationally costly. RFRBoost can reduce this by instead training random feature residual blocks and then fitting the comparatively lower-dimensional final representation to the final logits.
> > >
> > > **Time complexity correction:** In our previous reply, we unnecessarily simplified the training time complexity of RFRBoost. The detailed serial time complexity is $O(T(ND^2 + NDd + D^3 + dD^2 + Np^2 + NpD + p^3 + Dp^2))$, following Algorithm 2, but is highly parallelisable on GPU due to the heavy use of matrix multiplications. A more detailed breakdown will be provided in the revised manuscript.
> > >
> > > **Hyperparameters:** Unfortunately, we did not save the Optuna performance-over-trials data, only the final nested CV scores. The large number of hyperparameters and 100 Optuna trials per fold were primarily chosen for rigorous and fair comparison against XGBoost and especially E2E ResNets, which are notoriously sensitive and perform poorly without thorough tuning. Using at least 100 (optuna) random search trials is generally considered standard practice to our knowledge. As for RFRBoost, we generally found that the most important hyperparameter is the l2 regularization, and the number of boosting iterations. We found the other parameters (boosting learning rate, SWIM scale) to have a marginal effect on performance.

---

### Official Review · Reviewer_VGvF · 2025-03-12

**Overall Recommendation:** 4

**Summary:**

The paper introduces Random Feature Representation Boosting (RFRBoost), a novel approach that combines random feature neural networks (RFNNs) and gradient boosting theory to construct deep residual neural network models. The main idea is to build a deep ResNet structure using random feature layers that explicitly approximate the functional gradient of the network representation. This technique allows the resulting models to maintain computational efficiency while offering theoretical guarantees and improved performance.

**Claims And Evidence:**

Overall, the claims made in this papers are supported well. Yet, additional clarification (or empirical validation) may be beneficial:

1. Empirical validation primarily focuses on tabular datasets and one synthetic point-cloud dataset. This leaves unaddressed the applicability and performance on more structured or complex data (e.g., image, sequential, time-series).

2. The paper hypothesizes briefly that this improvement may arise due to the constraint on the functional gradient norm, but it lacks deeper analytical or empirical exploration of why this occurs.

3. While experimental evidence clearly supports improved computational efficiency relative to MLP ResNets, the scalability to larger or high-dimensional datasets remains unclear.

**Essential References Not Discussed:**

None.

**Experimental Designs Or Analyses:**

The experimental design and analyses presented in the paper were thoroughly reviewed for soundness and validity. Overall, the methodology employed by the authors is strong and convincing, particularly with their selection of datasets. The authors utilized a broad and diverse set of 91 benchmark tabular datasets sourced from OpenML, providing a comprehensive basis for evaluating the proposed method. Furthermore, they adopted a rigorous nested 5-fold cross-validation strategy, effectively minimizing biases related to hyperparameter tuning and yielding reliable estimates of model generalization. Bayesian optimization with Optuna for hyperparameter tuning further enhances the rigor of their evaluation process.

**Methods And Evaluation Criteria:**

Yes

**Other Comments Or Suggestions:**

None

**Other Strengths And Weaknesses:**

Although the method creatively combines random feature methods and gradient boosting, the primary conceptual ingredients—random features, residual blocks, and gradient boosting—are individually well-established. Thus, the originality of the contribution primarily arises from integration rather than fundamentally novel ideas.

The evaluation predominantly focuses on tabular datasets, limiting the significance of the method for broader machine learning contexts. Without evidence on structured datasets such as images, graphs, or sequential data, the broader significance of the proposed method remains unclear.

The surprising empirical advantage of gradient-greedy over exact-greedy approaches is not deeply analyzed theoretically or experimentally. Thus, despite its significance, the paper misses the opportunity to clarify why this approach works better, limiting conceptual clarity and interpretability.

**Questions For Authors:**

None

**Relation To Broader Scientific Literature:**

The key contributions of Random Feature Representation Boosting (RFRBoost) relate closely to existing literature on random feature neural networks (RFNNs), residual neural networks (ResNets), and gradient boosting theory. It integrates these approaches with gradient boosting concepts, which traditionally optimize ensembles of weak predictors (e.g., AdaBoost, Gradient Boosting Machines, XGBoost), but here, uniquely, optimize residual blocks within deep networks. Conceptually, the method builds upon recent theoretical advances that connect ResNets to gradient boosting and neural ordinary differential equations, generalizing these insights to deep RFNN architectures.

**Theoretical Claims:**

I did not check the correctness of the proofs in the Appendix. However, I have gone through the statement and claims in the main text of the paper, which looks good to me.

---

> ### Author Rebuttal · Authors · 2025-03-31
>
> We sincerely thank you for your thorough and positive assessment of our work and for your valuable comments. Below, we address your questions:
>
> **Empirical validation primarily focuses on tabular data:** The main question we set out to answer in our paper was whether one could build deep random feature neural networks (RFNNs) that significantly improve the performance of single-layer RFNNs while retaining their computational benefits. Since traditional RFNNs, using dense random projections, are primarily designed for tabular data, our paper focused on this domain. A careful study on tabular data is a necessary first step before looking at structured data like images or sequences, that would necessitate more complex random feature structures (e.g., random convolutional kernels or recurrent structures). Integrating such major architectural changes introduces complexities that make it difficult to address thoroughly within a single theory and within the scope of this single paper. We therefore consider this a valuable avenue for future research.
>
> **Q: The paper hypothesizes briefly that this improvement may arise due to the constraint on the functional gradient norm, but it lacks deeper analytical or empirical exploration of why this occurs.**
>
> We appreciate the request for further clarification, however we are not quite sure what "improvement" here is referring to. We give two possible interpretations:
>
> 1: Regarding the empirical observation in Table 1 where the gradient-greedy approach slightly outperforms the exact-greedy approach for regression tasks, we hypothesise that this could be due to an implicit regularisation effect. The gradient-greedy method, by focusing only on aligning with the functional gradient direction (under a norm constraint), might avoid overfitting compared to the exact-greedy method which directly minimizes the loss at each step. Providing a rigorous theoretical analysis of this phenomenon is challenging and would likely require fundamentally new tools, making it an interesting direction for future theoretical work.
>
>
> 2: Regarding why our gradient-greedy approach succeeds while Suggala et al. (2020) observed it performed worse for medium-sized SGD-trained networks, we refer to our discussion at the end of Section 2.2. The crucial difference lies in how the gradient step is implemented. Our gradient-greedy approach (Algorithm 2 and Theorem 3.2) explicitly solves an optimization problem constrained to find the unit-norm functional direction of the residual block $g$ that best aligns with the negative functional gradient $-\nabla \widehat{R}$ (see Theorem 3.2). This ensures that the learned residual block $g$ points in the correct direction in function space, allowing for a valid functional first-order approximation used in boosting (Eq. 4). In contrast, standard SGD training of a residual block to minimize the functional inner product $\langle g, \nabla_2 \widehat{R} \rangle_{L_2^D({\mu})}$ does not inherently enforce a constraint on the functional norm. We believe incorporating an explicit norm constraint, focusing on finding the optimal *direction*, is key to the success of the gradient-greedy strategy in our random feature setting.
>
>
>
> **Dataset Size and Scalability:** Please see `Dataset Size and Scalability' in response to Reviewer XEnQ.
>
>
> **Originality and Contribution**
> We respectfully disagree that our contribution is merely an integration of established components. While random features, residual blocks, and boosting are known concepts, successfully constructing *deep* RFNNs presents unique and significant challenges, since naively stacking random layers has been observed to degrade performance. Our key contribution is providing the first principled framework for building deep RFNNs using *any general* random feature type by rigorously using ideas from gradient representation boosting (which differs from classical gradient boosting). The use of random features within this framework is not just an implementation detail; it is fundamental to enabling the computational efficiency and theoretical analysis central to RFRBoost. This includes the closed-form solutions of Theorem 3.1, and the theory leading to the quadratically constrained least squares problem of Theorem 3.2.
>
>
>
>
>
> **Reviewer Comment: No supplementary material:**  We would like to clarify that supplementary material containing the Python code for our model implementations (RFRBoost, baselines) and experimental procedures was provided with our submission. We commit to making this code publicly available in a repository upon acceptance to ensure full reproducibility.
>
> ---
>
> We hope these clarifications adequately address your comments and further highlight the contributions of our work.

---

### Official Review · Reviewer_fxoT · 2025-03-12

**Overall Recommendation:** 3

**Summary:**

This paper studies the problem of boosting random features and its connection to ResNets. At stage $t$, a new group of random features $f_t = f_t(\Phi_{t-1})$ are generated and added to the previous stage's features $\Phi_t = \Phi_{t-1} + \Delta_t f_t$ where the matrix $\Delta_t$ gets optimized and the readout weights are also optimized. The authors study constraining $\Delta_t$ in various ways, optimization routines are derived for square and convex losses at the readout, and some theory is provided. In the last section, the method is compared to XGBoost, standard random feature methods, an end-to-end trained ResNet, and ridge regression applied to OpenML regression and classification tasks.

**Claims And Evidence:**

A number of tables are shown comparing the mean RMSE and fit times (across tasks) for the different methods. In general, the proposed method is close to as fast as XGBoost and shows similar performance. However, I have some issues with how the data are presented:
* Fig 2 and 3 "critical difference diagrams" are not discussed within the text as far as I can see. I am not familiar with this method of presenting data. This needs to be explained, since I don't know what I'm seeing here.
* Tables 1 and 2 are presented without any error bars on the RMSE or time numbers. The reader cannot evaluate whether these differences are significant or not. Since these numbers summarize performance over a large number of datasets (91 across both tables), we don't know how things vary here.
* Rather than summary tables like 1 & 2, it would be better to see the entire distribution of RMSE and fit time on all tasks. This could be shown with a plot of accuracy versus method with all datasets plotted as points and perhaps a box and whisker on top. I'm not familiar with how regression targets are generated in the OpenML datasets, but if these are of drastically different scales then summarizing this with mean RMSE will be misleading.
* The methods are meant to be scalable, however the authors restrict their study to datasets with less than 200 features and restrict themselves to 5000 observations per dataset. I'm not sure this is justified! If the fit times are as small as are reported, it seems reasonable to at least test a few larger datasets.

**Essential References Not Discussed:**

N/A

**Experimental Designs Or Analyses:**

Listed above in claims/evidence

How were the methods implemented?

**Methods And Evaluation Criteria:**

The method is only used with SWIM random features. These seem pretty specialized. How much do the results depend on this? These features depend on the previous layer features and data, but many people are interested in random feature models with, say, iid random Gaussian weights. Can you comment on whether the method works with these?

I think the OpenML datasets that are used seem fine. The point cloud separation task is pretty standard from the classification literature, but other "point cloud separation" data could be chosen as well.

I have listed other critiques of the evaluation above.

**Other Comments Or Suggestions:**

I have a number of small issues that I hope the authors will address here:
* Notation $\Delta_i$, $\partial_i$ is confusing and not standard to me. I believe the authors are indicating "in terms of the second argument" but should explain this notation.
* Line 111 right col: $W_t$ shape $C \times d$ isn't consistent with the $D \times d$ you use later.
* Lines 204-207 left col: Unclear what you mean by "diagonal matrix" for $Delta_t$ since it is rectangular. In general, how do $D$ and $d$ compare?
* Lines 232-233 left col: The language "sandwiched least squares" is evocative, but these solutions are known as "generalized Sylvester equation" solutions of the form $\sum_i A_i X B_i = C$ for unknown $X$. This could be worth mentioning.
* (throughout) I found the language "find amount of say" kind of distracting and unnecessary. I think you are just doing a line search; you might rephrase as "solve line search".

**Other Strengths And Weaknesses:**

My main concern with the paper is that I think the presentation of the experimental results is weak. If strengthened, I think the paper would be much better supported. Perhaps too much of the paper is taken up by Section 2 discussing all the various boosting methods.

**Questions For Authors:**

Given above

**Relation To Broader Scientific Literature:**

There seems to be a pretty good review of the existing literature, at least to my knowledge.

**Theoretical Claims:**

I checked the math for the least-squares problems that were solved for Theorem 3.1 I didn't find any issues. For Theorem 3.2, it could be simplifying to write the expression for the optimal $\Delta$ in terms of the pseudoinverse. If you change your definitions $F \to F^T$ and $G \to G^T$ then you get that
$$\Delta = \frac{\sqrt{n}}{ \| G \|_F} G F^\dagger$$
which is pretty evident from the minimization problem.

I'm not sure how useful the theoretical guarantee (Thm 3.4) is. I'm not sure if this is vacuous or not. How does this compare to other methods, for instance those in the Suggala et al (2020) paper that is referenced?

---

> ### Author Rebuttal · Authors · 2025-03-31
>
> We sincerely thank Reviewer fxoT for their careful reading and constructive feedback. Below, we provide detailed responses to each of the points raised.
>
> **On Critical Difference Diagrams:** To facilitate a meaningful comparison across all baseline models and RFRBoost variants, we use the Wilcoxon signed-rank test with Holm correction, as mentioned in Sec. 4.1 Evaluation Procedure. This statistical test assesses significant pairwise differences between models, and the average relative ranks and statistically significant groupings are presented visually in critical difference diagrams (Figures 2 and 3). If two models are connected by bars, then the difference between the two methods is not statistically significant. This is a well-established methodology for comparing multiple algorithms across multiple datasets [Demšar, 2006; Benavoli et al., 2016] and is standard practice within the broader statistical learning and data mining literature (see also the enthusiasm of Reviewer JjfD regarding the use of Wilcoxon signed-rank test). We aim to explain this in more detail in the revised manuscript.
>
> **Presentation of Results Tables:** We agree that presenting only mean scores limits the interpretation of the results. To address this, we will incorporated your suggestion of creating scatter plots overlaid with box-and-whiskers plots to visualize the distribution and variability of performance across all 91 datasets for each method. The critical difference diagrams (Figures 2 and 3) complement this by indicating which observed differences in relative rank are statistically significant. Furthermore, we will add a section to the Appendix with complete dataset-wise results for each model. All regression targets have been normalized to have 0 mean and variance 1 (Section 4.1), hence the results should be relatively comparable.
>
> **Dataset Size & Scalability:**
> Please see `Dataset Size and Scalability' in the response to Reviewer XEnQ.
>
> **SWIM initialization vs iid:** We performed experiments using both iid random features and SWIM initialized random features. We initially presented only the SWIM results in the main paper due to slightly superior empirical performance and the 8 page space constraint. In the revised manuscript, we will add an ablation study with pairwise plots directly comparing RFRBoost with iid features versus SWIM features, quantifying the (small) difference in performance. For example, SWIM features only beat iid features 51 out of 91 times.
>
> **How were the methods implemented?** We will add this missing detail to the revised manuscript. All baseline models were implemented in PyTorch, and RFRBoost in particular was implemented following Algorithms 1 and 2. We used LBFG-S as the minimizer for log likelihood loss, and standard torch linear algebra routines for the (sandwiched) least squares, following our derived formulas in the Appendix. E2E ResNets were trained with Adam as described in Section 4.1. The full code is currently contained in the supplementary material and we will additionally release it in a public repository upon acceptance.
>
> **Pseudoinverse:** Thanks for this remark. We will point out the alternative expression in terms of the pseudoinverse in the revised version.
>
> **Theoretical Regret Bound:** Our regret bound in Theorem 3.4 is of the same order as similar results in the literature, e.g. Suggala et al (2020) Corollary 4.3. We will make this clearer in the revised manuscript.
>
> Minor comments:
>
> **Notational confusion: $\nabla_i, \Delta_i, \partial_i$:**
> Our original motivation for using $\Delta_t$ stemmed from the connection to the discrete time step in the Euler discretization view of ResNets (Eq. 1). Recognizing the ambiguity, we have replaced $\Delta_t$ (the linear map within the residual block) with $A_t$ throughout the paper. We have also ensured that the text in Section 2 and Appendix B/C clearly specifies when partial derivatives or gradients are taken with respect to specific function arguments.
>
> **Line 111 $W_t$ shape:** Thanks for spotting this. We will correct the dimensions in the revised manuscript.
>
> **Lines 204-207 'diagonal $\Delta_t$':** We have improved the text to make it clearer that we only consider $\Delta_{\text{diag}}$ and  $\Delta_{\text{scalar}}$ in the case when hidden size is equal to feature dimension size $D=p$. The dense case is free of this restriction, allowing for rectangular matrices.
>
> **Sandwiched Least Squares:** We have modified the paper to properly address the 'sandwiched least squares' problems as Generalized Sylvester Equations.
>
> **Usage of 'amount of say':** While "amount of say" is standard terminology used in the classical boosting literature, notably in AdaBoost [Freund & Schapire, 1997], we understand it may be unfamiliar to a broader audience. We have adopted your suggestion and replaced it with "line search step size" or similar phrasing for improved clarity.
>
> ---
>
> We hope these responses and revisions address your concerns adequately.

---

> > ### Comment · Reviewer_fxoT · 2025-04-02
> >
> > The authors have addressed my concerns in a good way. I look forward to seeing the more detailed experimental results. Will it be possible to see these in a preliminary revision before I finalize my score?

---

> > > ### Author Response · Authors · 2025-04-08
> > >
> > > Thank you for your response.
> > >
> > > Some preliminary larger-scale experiments have now been provided in our response to Reviewer XEnQ.
> > >
> > > Additionally, linked below are the two requested box-and-whiskers plots overlaid with scatter points:
> > >
> > > Regression: https://postimg.cc/Q98d0Y9d
> > >
> > > Classification: https://postimg.cc/tsvkX8bL
> > >
> > > Due to the large number of diverse datasets, these plots unfortunately become quite dense and potentially less readable and informative than hoped. Therefore, while we will include them in the revision, we still believe the rigorous critical difference diagrams (Figures 2 and 3), combined with the full dataset-wise results tables (which we will add to the Appendix), offer a clearer and statistically more informative summary of the comparative results.
> > >
> > > ---
> > >
> > > We hope that providing these additional plots and results, alongside addressing your previous comments, proves helpful for your final assessment and score.

---

### Decision · Program_Chairs · 2025-05-01

**Decision:**

Accept (poster)

**Comment:**

## Summary
The paper introduces Random Feature Representation Boosting (RFRBoost), a method that combines random feature neural networks and gradient boosting theory to construct deep residual neural network models. It outperforms traditional RFNNs and end-to-end trained ResNets, and improves computational efficiency and performance compared to XGBoost and other methods.

## Strengths
- Utilized 91 benchmark tabular datasets from OpenML for comprehensive evaluation.
- Adopted a rigorous nested 5-fold cross-validation strategy to minimize hyperparameter tuning biases.
- RFRBoost integrates existing literature on RFNNs, ResNets, and gradient boosting theory.
- Uniquely optimizes residual blocks within deep networks, enhancing model generalization.
- Builds upon recent theoretical advances connecting ResNets to gradient boosting and neural ordinary differential equations.

## Weaknesses
- The paper's experimental results presentation is weak.
- The paper's section on various boosting methods is overly extensive.
- The method combines random feature methods and gradient boosting, but its originality comes from integration rather than novel ideas.
- The evaluation primarily focuses on tabular datasets, limiting its significance for broader machine learning contexts.
- The paper fails to analyze the empirical advantage of gradient-greedy over exact-greedy approaches, limiting conceptual clarity and interpretability.

## Conclusions

Based on the reviews, the author feedback and postrebuttal reviews, the paper is a weak accept for ICML since the experimental validation is the weakest part of the paper. The paper should include more experiments since the new figures provided does not show clearly an improvement over the baselines